# ClusCAM: Clustered Visual Explanations for Vision Models in Image Classification

## Abstract

As deep neural networks continue to achieve considerable success in high-stakes computer vision applications, the demand for transparent and interpretable decision-making is becoming increasingly critical. Post-hoc explanation methods, such as Class Activation Mapping (CAM), were developed to enhance interpretability by highlighting important regions in input images. However, existing methods often treat internal representation (feature maps or patch tokens) as independent and equally important, neglecting their semantic interactions, which can result in irrelevant or noisy signals in the explanation. To overcome these limitations, we propose ClusCAM, a gradient-free post-hoc explanation method that groups internal representations into meaningful clusters, referred to as meta-representations. We then quantify their importance using logit differences with discarding and temperature-scaled softmax to focus on the most influential groups. By modeling group-wise interactions, ClusCAM produces sharper and more interpretable explanations. The approach is architecture-agnostic and applicable to both Convolutional Neural Networks and Vision Transformers. Through our extensive experiments, ClusCAM outperforms the state-of-the-art methods by up to 17.8% and 24.19% improvement in Increase in Confidence and Average Gain, respectively, and produces visualizations more faithful to the model's prediction.

## 1 Introduction

Deep vision models, such as Convolutional Neural Networks (CNNs) and Vision Transformers (ViTs), have become the foundation of modern image classification systems. However, they are often criticized as "black boxes" due to their lack of interpretability: it remains unclear which internal representations drive specific decisions, making these models difficult to trust and analyze in critical applications (Bharati et al., 2023; Belharbi et al., 2022). The need to assess model behavior, therefore, has led to the development of eXplainable Artificial Intelligence (XAI) techniques, particularly post-hoc explanation methods. Among these, Class Activation Mapping (CAM) represents a foundational line of work that generates class-specific saliency maps by linearly combining activation maps, typically, from the final convolutional layer in CNNs (Zhou et al., 2016). These maps highlight spatial regions in the input image that most contribute to the model's prediction. Due to its architectural simplicity and extensibility, CAM has become a standard baseline for explaining CNNs and has been extended to ViTs in recent works (Zhang et al., 2024; Wu et al., 2024).

Over the years, CAM-based methods have evolved into two main groups: *gradient-based* and *gradient-free* approaches. Gradient-based methods, such as GradCAM (Selvaraju et al., 2016) and GradCAM++ (Chattopadhay et al., 2018), compute the gradients of the target output with respect to intermediate feature maps, thereby estimating which activations have the strongest influence on the prediction. In contrast, gradient-free methods, including ScoreCAM (Wang et al., 2020), AblationCAM (Ramaswamy et al., 2020), ReciproCAM (Byun & Lee, 2024), OptiCAM (Zhang et al., 2024), and ShapleyCAM (Cai, 2025), avoid gradient computations by masking or perturbing the feature maps and directly observing the impact on the model output. However, most existing methods treat internal representations (e.g., activation maps or patch tokens) as independent and equally important, ignoring possible interactions and their collective contributions. This may lead to less reliable saliency maps, limiting interpretability. To overcome these shortcomings, we introduce ClusCAM, a novel post-hoc explainability method that clusters internal representations into similar groups called *meta-representations* and attributes class-specific importance to them based on logit

differences. This group-wise modeling captures high-level interactions among features and filters out irrelevant grouped components, as illustrated in Fig. 1.

Following other state-of-the-art (SoTA) CAM-based methods in the literature, ClusCAM is evaluated on the ILSVRC benchmark (Russakovsky et al., 2015). We also further validate the effectiveness of ClusCAM in healthcare through a real-world Alzheimer's disease dataset (Falah.G.Salieh, 2023). Quantitative results coupled with qualitative visualizations demonstrate that ClusCAM provides explanations that are more interpretable and better aligned with the model's predictions. In summary, our key contributions are as follows:

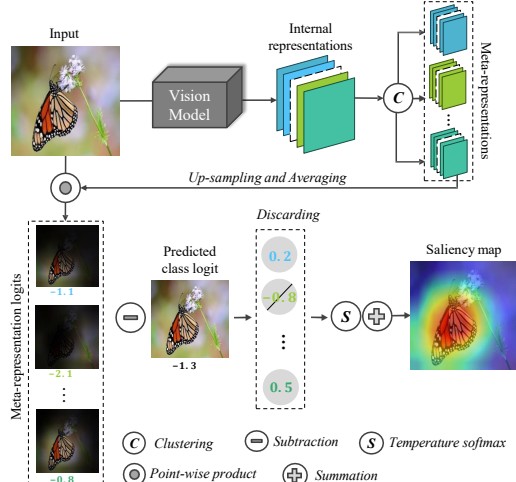

- We propose *ClusCAM*, a gradient-free method that overcomes the limitations of current methods that treat internal representations independently and equally (Sec. 3).

- We introduce a principled procedure for selecting key hyperparameters based on validation dynamics, curvature analysis, and probabilistic modeling, eliminating the need for manual tuning (Sec. 3.4).

- We empirically demonstrate that ClusCAM significantly outperforms SoTAs in terms of interpretability and faithfulness across various architectures and multiple metrics.(Sec. 4).

Figure 1: Overview of ClusCAM: Internal representations are clustered into meta-representations, each of which masks the input to obtain a logit. The scores, computed as the logit differences from the predicted class logit with discarding and temperature scaling, serve as respective weights. Then, a score-weighted summation of the meta-representations yields the final saliency map.

## 2 RELATED WORK

CAM (Zhou et al., 2016) is a prevalent approach to interpret how vision models predict from input images thanks to its intuitive mechanism. Given a CNN containing a Global Average Pooling (GAP) layer between the last convolution layer and the last Fully Connected (FC) layer, for a target class $c$, the CAM explanation is defined as follows:

$$E_{\text{CAM}}^c = \sigma(\sum_i \alpha_i^c A^i), \tag{1}$$

where $\alpha_i^c$ denotes the weight of the $i$-th neuron after GAP, $A^i$ is the $i$-th feature map, and $\sigma$ represents the ReLU function. Although CAM has limited flexibility due to its constraints with architectures with a GAP layer followed by an FC classifier (He et al., 2022), it has laid a foundation for subsequent studies in the domain. Typically, these works can be categorized into two main groups: gradient-based and gradient-free methods.

### 2.1 GRADIENT-BASED METHODS

Gradient-based methods score the importance of each feature map using integrated gradients and can be applied to any classification architecture based on backpropagation. Selvaraju et al. (2016) extended the original CAM to GradCAM by incorporating gradients from any target class into the last convolutional layer, which is formulated as:

$$E_{\text{Grad}}^c = \sigma\left(\frac{1}{Z} \sum_i \sum_{u,v} \frac{\partial y^c}{\partial A^i(u,v)} A^i\right), \tag{2}$$

where $Z$ is the number of pixels in feature map $A^i$, $y^c$ is the logit (pre-softmax output) for class $c$, and $A^i(u,v)$ represents the pixel at $(u,v)$ in $A^i$.

Building on the same principle of using partial derivatives, later methods such as GradCAM++ (Chattopadhay et al., 2018) and XGradCAM (Fu et al., 2020) refine the computation of importance weights to enhance visualization precision and stability, while providing more flexibility for interpreting

CNNs. Likewise, other methods like LayerCAM (Jiang et al., 2021) and GroupCAM (Zhang et al., 2021) still rely on gradients but incorporate additional information from the CNN itself.

Besides, all gradient-based methods are constrained in *post-deployment settings* (e.g., ONNX (Bai et al., 2019) or OpenVINO (Intel, 2019)) with frozen model weights. Additionally, Wang et al. (2020) have identified two more drawbacks of such approaches: *saturation*, where gradients can become noisy or vanish due to non-linearities; *false confidence*, where feature maps with high weights may contribute little to the model's output. These issues highlight the need for gradient-free methods.

## 2.2 GRADIENT-FREE METHODS

Gradient-free methods estimate feature importance through the effect of masked or ablated feature maps on the model's output. Among the SoTAs, ScoreCAM (Wang et al., 2020) generates explanations (saliency maps) by masking the input with upsampled feature maps and measuring the change in the model's output relative to a baseline:

$$E_{\text{Score}}^c = \sigma \left( \sum_i \text{softmax} \left( y^c(x_i') - y^c(x_b) \right) A^i \right),$$ (3)

where $x_b$ is the baseline image, and $x_i' = x \odot \texttt{NormalizeUpsample}(A^i)$, with $\odot$ denoting the point-wise product. Another approach, AblationCAM (Ramaswamy et al., 2020), estimates feature-map importance by quantifying the change in prediction upon its removal. More fine-grained approaches, including ReciproCAM (Byun & Lee, 2024) and ShapleyCAM (Cai, 2025), apply pixel-level masking across all feature maps to assess importance. However, these methods may yield fragmented saliency maps when they fail to capture the broader semantic context through pixel relationships. Notably, these methods have primarily been designed and validated on CNNs, with limited evidence of their applicability and effectiveness on ViTs. Addressing this gap, OptiCAM (Zhang et al., 2024) generalizes CAM to ViTs by extending feature-map combination from a linear to a non-linear formulation via optimized contribution weights. Unlike prior works, OptiCAM is benchmarked on ViTs, where it exhibits superior performance over earlier CAM variants, underscoring the need for methods that generalize across both CNNs and ViTs.

## 2.3 EXPLAINABILITY FOR ViTs

Explainability for ViTs has recently attracted increasing attention, as CNN-based explanation methods may not directly transfer to token-based architectures with global self-attention. Chefer *et al.* (Chefer et al., 2021) showed that attention visualization alone is insufficient and proposed a relevance propagation framework combining attention and gradient signals. In addition, hybrid strategies that leverage both forward attention and backward gradients have been explored to suppress noise in transformer explanations, such as AG-CAM (Leem & Seo, 2024). More recent studies highlight the importance of token transformations and aggregation when explaining ViTs (Wu et al., 2024; Bousselham et al., 2024), showing that ignoring token interactions can lead to misleading or incomplete explanations.

**Limitations of existing CAM-based methods.** Despite their differences, most CAM-based approaches share several common limitations. First, they typically assign importance scores to individual representations (feature maps or patch tokens), implicitly assuming their independence, and thus overlook group-level interactions where multiple representations jointly encode higher-level semantics (Stone et al., 2017; Zeiler & Fergus, 2014). Second, they often treat all internal representations equally during aggregation, ignoring their heterogeneous contributions to a specific prediction (Zimmermann et al., 2021), which can introduce irrelevant or noisy explanations. Finally, although recent works extend CAM to ViTs, most evaluations focus only on the original ViT architecture, with limited validation on its diverse variants (e.g., DeiT, Swin). This lack of cross-architecture analysis raises concerns about the generalizability of current explanation methods and highlights the need for a more architecture-agnostic explanation mechanism that can be applied consistently to both CNNs and ViTs.

## 3 PROPOSED METHOD

We propose ClusCAM, a gradient-free visual explanation method that accounts for feature interactions and aligns importance attribution. The core idea is to cluster internal representations into similarity groups (meta-representations) and quantify their contribution to the model's prediction. Fig. 1 provides an overview of the overall pipeline, while the following subsections elaborate on its components in more detail.

### 3.1 INTERNAL REPRESENTATION GROUPING

Prior studies have shown that individual feature maps often correspond to low-level or mid-level patterns, while their combinations capture higher-level visual patterns (Zeiler & Fergus, 2014; Bau et al., 2017; Panousis & Chatzis, 2023). Therefore, treating each representation independently, as commonly done in existing CAM-based methods, often overlooks possible cooperative interactions, leading to fragmented or noisy explanations. Inspired by this observation, we propose to group co-activated internal representations into meta-representations, which better reflect the spatial dependencies and collective contribution to the model's decision.

We now describe how ClusCAM constructs these meta-representations. Consider a pre-trained vision model (CNN or ViT) with an input image $x \in \mathbb{R}^{H \times W \times D}$. For CNNs, the output of the last convolutional layer is a set of $N$ feature maps, $\mathcal{F} = \{F_1, \ldots, F_N\}$, where each $F_i \in \mathbb{R}^{p \times q}$ encodes localized patterns. For ViTs, the representation is the patch token.

First, the co-activated representations are clustered into partition $\mathcal{F}$ into $K$ disjoint groups $\mathcal{G} = \{\mathcal{G}_1, \mathcal{G}_2, \ldots, \mathcal{G}_K\}$, such that $\bigcup_j \mathcal{G}_j = \mathcal{F}$ and $\mathcal{G}_i \cap \mathcal{G}_j = \emptyset$. In each group $\mathcal{G}_j$ that captures a set of co-activated patterns in the input, a meta-representation $\mathcal{M}_j$ is defined as a group representation:

$$\mathcal{M}_j = \frac{1}{|\mathcal{G}_j|} \sum_{F \in \mathcal{G}_j} \texttt{NormalizeUpsample}(F), \tag{4}$$

where $\texttt{Normalize}(.)$ denotes a linearly scaling normalization that maps each element into the range $[0; 1]$ and $\texttt{Upsample}(.)$ is a bilinear interpolation operation that resizes $F$ into the input size.

### 3.2 SCORING VIA LOGIT DIFFERENCES

Intuitively, each meta-representation $\mathcal{M}_j$ highlights a spatial region corresponding to a group of co-activated patterns. To quantify how much this group contributes to the model's prediction, we isolate the region it emphasizes and observe the resulting variation in the model's output.

Specifically, let $f_{\text{logit}}(x) \in \mathbb{R}^C$ denote the model's output logits over $C$ classes, and let $c \in \{1, \ldots, C\}$ be the target class. The importance of $\mathcal{M}_j$ is assessed by measuring the change in class logit when only the regions emphasized by $\mathcal{M}_j$ are retained in the input image. The importance score of $\mathcal{M}_j$ is then defined as the logit difference:

$$s_j^c = f_{\text{logit}}^c(x \odot \mathcal{M}_j) - f_{\text{logit}}^c(x), \tag{5}$$

where $\odot$ denotes element-wise product.

### 3.3 DISCARDING AND SOFTMAX-BASED REFINEMENT

Not all meta-representations contribute positively to the prediction, as some may capture spurious patterns that distract the model and reduce class logit (c.f. empirical example in the Appendix D). To suppress such possible effects, ClusCAM filters out noisy groups using a discarding mechanism and temperature-scaled weighting. Specifically, we discard the $r\%$ least important meta-representations, ranked by their scores $s_j^c$, and retain a subset $\mathcal{S} \subset \{1, \ldots, K\}$ of the most influential ones.

To combine the retained meta-representations into a final saliency map, we normalize their scores using a temperature softmax with the parameter $\tau \in (0; 1)$ that controls the sharpness of the distribution:

$$\alpha_j = \frac{\exp(s_j^c / \tau)}{\sum_{k \in \mathcal{S}} \exp(s_k^c / \tau)}, \quad j \in \mathcal{S}, \tag{6}$$

This helps highlight salient regions and suppress less relevant ones in the final explanation visualization, making it a more focused and interpretable. Finally, the class-specific saliency map is computed as a weighted sum of the selected meta-representations:

$$E_{\text{Clus}}^c = \texttt{Normalize}(\sum_{j \in \mathcal{S}} \alpha_j \cdot \mathcal{M}_j). \tag{7}$$

In summary, three stages of ClusCAM jointly enable the generation of faithful and focused saliency maps. By clustering representations into meta-representations, quantifying their class relevance through logit differences, and discarding out spurious groups via discarding and temperature softmax, ClusCAM yields structured visualizations that better align with the model's behavior.

## 3.4 HYPERPARAMETER SELECTION

In this section, we describe how the key hyperparameters of ClusCAM are determined in a principled and data-driven manner, including the clustering strategy, the number of groups $K$, the discarding ratio $r$, and the temperature $\tau$.

**Clustering algorithm.** We adopt K-means++ clustering due to its simplicity, fast convergence, and suitability for grouping representations based on activation similarity. Since our goal is to cluster feature maps (or tokens) based on their co-activation patterns, K-means++ provides an effective choice by minimizing intra-cluster variance in the feature space. Moreover, it does not require additional supervision or model retraining. Empirically, K-means++ is also deemed an effective choice for constructing meta-representations in ClusCAM (c.f. Section 4.4).

**The number of groups $K$.** We determine $K$ using a data-driven Elbow criterion based on curvature analysis (Bholowalia & Kumar, 2014). In particular, we define a proxy function $P(K)$ over a held-out validation set:

$$P(K) = \frac{1}{K|V|} \sum_{x \in V} \sum_{i=1}^{K} \left( f_{\text{logit}}^c(x^{(i)}) - f_{\text{logit}}^c(x) \right),$$ (8)

where $V$ is the validation set.

**Estimating $r$.** The discarding ratio $r \in (0, 1)$, representing the fraction of discarded groups, is estimated via a two-component Gaussian Mixture Model over group importance scores:

$$r = \frac{1}{K|V|} \sum_{i=1}^{K|V|} \mathbb{P}(z_i = \text{non-salient} \mid s_i),$$ (9)

where $z_i$ is the latent group assignment and $s_i$ its importance.

**Setting $\tau$.** The temperature $\tau$ controls the sharpness of importance weights across selected groups. We define:

$$\tau = \frac{1}{\log(1 + rK)}.$$ (10)

This adaptive scaling ensures sharper distributions when more groups are retained. It also guarantees $\tau > 0$ for all valid $r, K$, avoiding negative or undefined temperatures.

The completed procedure of ClusCAM is presented in Alg. 1, the details of hyperparameter selections, including two algorithms for selecting $K$ and $r$, as well as a sensitive analysis in the Appendix B.

## 4 EXPERIMENT

Our experimental analysis is organized into four parts. First, we show that meta-representations can increase model logits (confidence). Second, we provide a quantitative evaluation using three standard metrics to benchmark ClusCAM against the seven most common CAM-based approaches. Next, we perform a qualitative assessment of explanation quality under different visual scenarios. Finally, we do the ablation study to understand the impact of each component in our design.

**Datasets**. Following other baseline methods in the domain, we use the ILSVRC2012 benchmark (Russakovsky et al., 2015) for natural images. We also employ the Alzheimer's MRI dataset (Falah.G.Salieh, 2023) to evaluate ClusCAM in medical imaging applications.

**Network architectures**. We employ widely-used models in image classification, including CNNs such as the ResNet family (ResNet-18/34/50/101), EfficientNet, and InceptionNet, as well as ViTs like ViT-B, Swin-B, LeViT-192/256, CaiT-XXS-24, and PVTv2.

More details about the experimental implementation can be found in the Appendix A, the complete code is provided in the Supplementary Materials.

### 4.1 EFFECT OF META-REPRESENTATIONS ON MODEL LOGITS

To evaluate the impact of meta-representations, we compare them against the baseline obtained by averaging the internal representations. Both approaches produce cluster-level logits for the same set of 2,000 samples from the ILSVRC dataset. Fig. 2 clearly illustrates that meta-representations

---

**Algorithm 1** ClusCAM Algorithm

---

**Input:** Image $x$, trained vision model $f$, target class $c$, number of groups $K$, discarding ratio $r$, temperature $\tau$.

**Output:** $E_{\text{Clus}}^c$, saliency map for class $c$.

**Procedure:**

1: Extract internal representations $\mathcal{F} = \{F_1, \ldots, F_N\}$ from $f(x)$;
2: Flatten each $F_i \in \mathbb{R}^{p \times q}$ into a $p \times q$ vector;
3: Cluster $\mathcal{F}$ into $K$ disjoint groups $\{\mathcal{G}_1, \ldots, \mathcal{G}_K\}$ using K-Means++;
4: **for all** group $\mathcal{G}_j$ **do**
5: $\quad \mathcal{M}_j \leftarrow \frac{1}{|\mathcal{G}_j|} \sum_{F \in \mathcal{G}_j} \texttt{NormalizeUpsample}(F)$;
6: **end for**
7: **for all** meta-representation $\mathcal{M}_j$ **do**
8: $\quad$ Generate masked input: $x^{(j)} \leftarrow x \odot \mathcal{M}_j$;
9: $\quad$ Compute importance: $s_j^c \leftarrow f_{\text{logit}}^c(x^{(j)}) - f_{\text{logit}}^c(x)$;
10: **end for**
11: : $\mathcal{S} \leftarrow$ remove the bottom $r\%$ lowest-scoring groups in $\{s_j^c\}$;
12: **for all** $j \in \mathcal{S}$ **do**
13: $\quad \alpha_j \leftarrow \frac{\exp(s_j^c/\tau)}{\sum_{k \in \mathcal{S}} \exp(s_k^c/\tau)}$;
14: **end for**
15: Compute saliency map and normalize to $[0, 1]$:
16: $\quad E_{\text{Clus}}^c \leftarrow \texttt{Normalize}\left(\sum_{j \in \mathcal{S}} \alpha_j \cdot \mathcal{M}_j\right)$;
17: **return** $E_{\text{Clus}}^c$.

---

yield higher logits. The boxplots confirm that this effect holds for each group, while the histogram shows the global distribution of differences shifted far to the positive side. The statistical tests in Tab. 1 further support this observation: Both parametric and non-parametric tests strongly reject $H_0$ (all one-sided; $p < 10^{-199}$), and the effect sizes are uniformly large ($d \approx 0.82$), providing strong evidence that meta-representations significantly increase model logits compared to the baseline.

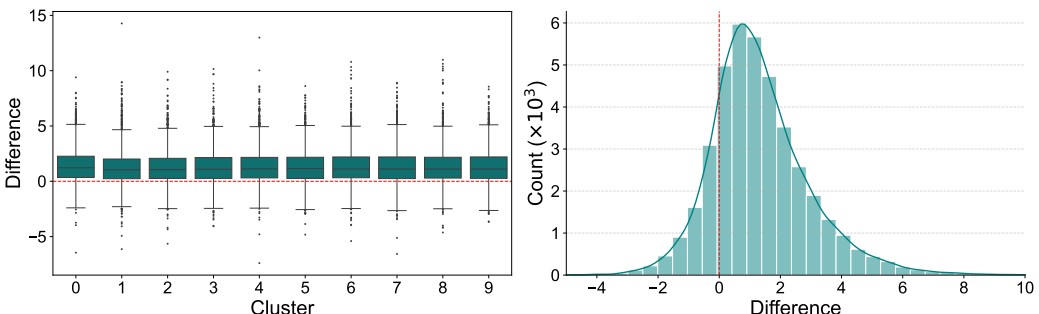

Figure 2: Comparison of model outputs obtained with meta-representations versus the baseline. (**Left**) Boxplots: group-wise differences for the top 10 groups. (**Right**) Histogram: overall distribution of differences across all groups and samples.

Table 1: Statistical summary. We report average logit differences between meta-representations and the baseline, effect sizes (Cohen's $d$), and $p$-values from a one-sided paired t-test ($H_0 : \mu \leq 0$) and a one-sided Wilcoxon signed-rank test ($H_0 :$ median $\leq 0$).

| | Mean Difference (range) | Cohen's $d$ (range) | $p$ (t-test) | $p$ (Wilcoxon) |
|---|---|---|---|---|
| Across all clusters | $1.31 \pm 0.05$ (1.25–1.41) | $0.82 \pm 0.04$ (0.76–0.88) | $< 10^{-202}$ | $< 10^{-199}$ |

## 4.2 QUANTITATIVE ANALYSIS

The quantitative evaluation is conducted using three widely-used metrics: Average Drop (AD) (Chattopadhay et al., 2018) measures the reduction in prediction confidence when only the most salient regions are retained. Increase in Confidence (IC) (Chattopadhay et al., 2018) measures the proportion

Table 2: Evaluation of various CAM-based approaches on the ILSVRC and Alzheimer's datasets, averaged over 6 CNNs and 6 ViTs. AD: Average Drop; IC: Increase in Confidence; AG: Average Gain; ↓ / ↑: lower/higher is better. The best is highlighted in **bold** while the second rank is in *italics*.

| ILSVRC | Metric | GradCAM | GradCAM++ | ScoreCAM | AblationCAM | ReciproCAM | OptiCAM | ShapleyCAM | ClusCAM |
|---|---|---|---|---|---|---|---|---|---|
| CNNs | AD (↓) | $18.93 \pm 4.82$ | $20.00 \pm 6.47$ | $14.66 \pm 9.25$ | $18.99 \pm 4.87$ | $23.59 \pm 6.36$ | *$8.75 \pm 2.08$* | $18.59 \pm 4.84$ | **$7.82 \pm 2.40$** |
| | IC (↑) | $35.07 \pm 4.62$ | $33.35 \pm 5.59$ | *$47.99 \pm 9.88$* | $34.68 \pm 4.51$ | $30.95 \pm 4.74$ | $41.78 \pm 3.42$ | $35.32 \pm 5.29$ | **$59.58 \pm 5.82$** |
| | AG (↑) | $17.25 \pm 4.34$ | $15.87 \pm 4.66$ | *$26.15 \pm 9.60$* | $14.21 \pm 5.19$ | $15.42 \pm 3.93$ | $13.90 \pm 4.96$ | $15.26 \pm 6.34$ | **$33.10 \pm 8.01$** |
| ViTs | AD ↓ | $76.68 \pm 14.01$ | $70.53 \pm 13.03$ | $56.21 \pm 26.64$ | $65.19 \pm 17.36$ | $40.46 \pm 12.86$ | **$4.12 \pm 1.72$** | $74.58 \pm 18.36$ | *$5.64 \pm 4.07$* |
| | IC ↑ | $4.52 \pm 3.27$ | $4.95 \pm 3.19$ | $14.76 \pm 12.20$ | $7.92 \pm 5.24$ | $10.39 \pm 4.81$ | *$41.28 \pm 10.93$* | $5.03 \pm 4.43$ | **$54.71 \pm 21.53$** |
| | AG ↑ | $1.61 \pm 1.47$ | $1.81 \pm 1.77$ | $6.93 \pm 5.66$ | $5.94 \pm 8.07$ | $4.80 \pm 2.66$ | *$9.00 \pm 7.35$* | $1.13 \pm 0.52$ | **$31.22 \pm 23.91$** |

| Alzheimer's | Metric | GradCAM | GradCAM++ | ScoreCAM | AblationCAM | ReciproCAM | OptiCAM | ShapleyCAM | ClusCAM |
|---|---|---|---|---|---|---|---|---|---|
| CNNs | AD (↓) | $17.92 \pm 20.00$ | $17.12 \pm 18.57$ | $13.87 \pm 17.32$ | $16.34 \pm 20.51$ | $17.71 \pm 18.81$ | **$9.51 \pm 19.51$** | $18.18 \pm 20.54$ | *$11.25 \pm 14.95$* |
| | IC (↑) | $32.02 \pm 24.65$ | $33.06 \pm 23.66$ | $41.96 \pm 27.24$ | $34.48 \pm 23.07$ | $31.68 \pm 25.17$ | *$49.60 \pm 19.90$* | $31.59 \pm 25.12$ | **$65.00 \pm 21.97$** |
| | AG (↑) | $32.55 \pm 26.48$ | $33.76 \pm 26.84$ | *$42.71 \pm 27.38$* | $34.77 \pm 24.86$ | $32.14 \pm 27.17$ | $34.03 \pm 17.93$ | $32.12 \pm 26.51$ | **$58.22 \pm 17.42$** |
| ViTs | AD (↓) | $49.58 \pm 33.32$ | $45.50 \pm 27.51$ | $39.36 \pm 20.98$ | $47.67 \pm 33.24$ | $40.55 \pm 39.33$ | *$8.93 \pm 11.36$* | $47.40 \pm 31.98$ | **$8.79 \pm 8.95$** |
| | IC (↑) | $16.41 \pm 21.05$ | $14.87 \pm 18.62$ | $23.81 \pm 24.04$ | $14.02 \pm 21.85$ | $22.28 \pm 22.09$ | *$46.30 \pm 23.16$* | $16.33 \pm 19.69$ | **$55.24 \pm 21.62$** |
| | AG (↑) | $8.81 \pm 17.54$ | $8.25 \pm 15.77$ | *$11.41 \pm 20.59$* | $7.97 \pm 16.92$ | $7.76 \pm 14.88$ | $9.96 \pm 11.66$ | $8.60 \pm 16.95$ | **$17.44 \pm 21.15$** |

of samples for which the model's confidence increases when restricting the input to the highlighted regions. Average Gain (AG) (Zhang et al., 2024) reports the average change in confidence score across all masked inputs. Unlike IC, which focuses on frequency, AG quantifies the magnitude of confidence improvement. Formal definitions of these metrics, along with additional analyses on the localization ability of explanations (i.e., how well the highlighted regions align with the true object of interest), are presented in the Appendix C and D.2, respectively. Moreover, an additional experiment with a ViT-specific baseline is provided in D.3. Tab. 2 summarizes the results across all evaluated metrics, with the best and second-best scores highlighted per metric and architecture.

**On CNNs,** ClusCAM outperforms all existing CAM-based methods across both datasets, achieving substantial improvements with large margins of 17.8% in IC and 24.19% in AG compared to the second-best approach, except for a slight degradation (1.74%) in AD on the Alzheimer's dataset.

**On ViTs,** ClusCAM surpasses all baselines across both datasets, with large margins of 13.43% in IC and 22.22% in AG on the ILSVRC dataset, at the cost of a slight degradation of 1.52% in AD. On the Alzheimer's dataset, it consistently achieves the best results across all three metrics, highlighting its strong effectiveness on transformer architectures.

### 4.3 QUALITATIVE ANALYSIS

To reflect the spatial quality of saliency maps, we qualitatively evaluate how well different methods localize class-relevant regions under three settings as suggested by Byun & Lee (2024), including (i) single-object, (ii) multiple objects of the same class, and (iii) multiple objects with different classes.

**Explanation for CNNs.** Fig. 3 summarizes the qualitative comparison of CAM-based methods across three scenarios. In the single-object case (first row), most methods emphasize the head region, while ScoreCAM, OptiCAM, and ClusCAM additionally capture the tail, with ClusCAM highlighting both the tail and the supporting branch more distinctly. For multiple objects of the same class (second row), several baselines tend to focus on a single dominant instance, whereas ScoreCAM, OptiCAM, and ClusCAM succeed in highlighting both. In the different-class setting (last row), GradCAM++ and OptiCAM perform poorly, as their saliency maps are either scattered or unfocused, whereas the remaining methods deliver more accurate and localized explanations. Overall, ClusCAM and ScoreCAM consistently produce sharper and more comprehensive explanations across the three scenarios.

**Explanation for ViTs.** In Fig. 4, methods such as GradCAM, GradCAM++, ReciproCAM, and ShapleyCAM tend to highlight only a few sparse and scattered regions, failing to capture the overall object structure, while the remaining methods activate broader areas. Specifically, ScoreCAM and AblationCAM often emphasize background regions rather than the object itself. In contrast, both OptiCAM and ClusCAM successfully focus on the object, but ClusCAM produces more complete and coherent explanations, better aligning with object boundaries. These qualitative results match the quantitative improvements reported in Tab. 4 in the Appendix D, where ClusCAM consistently achieves the lowest localization error compared to all baselines.

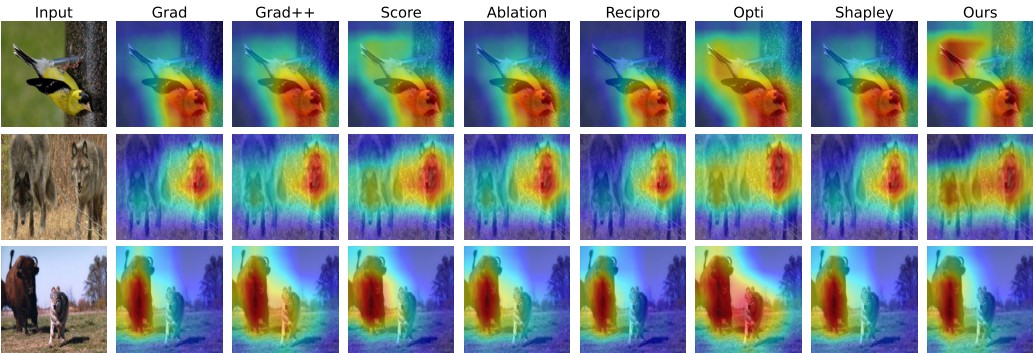

Figure 3: Visual explanations generated by various CAM-based methods for ResNet-18, from top to bottom: single-object, multiple objects of the same class, and multiple objects with different classes.

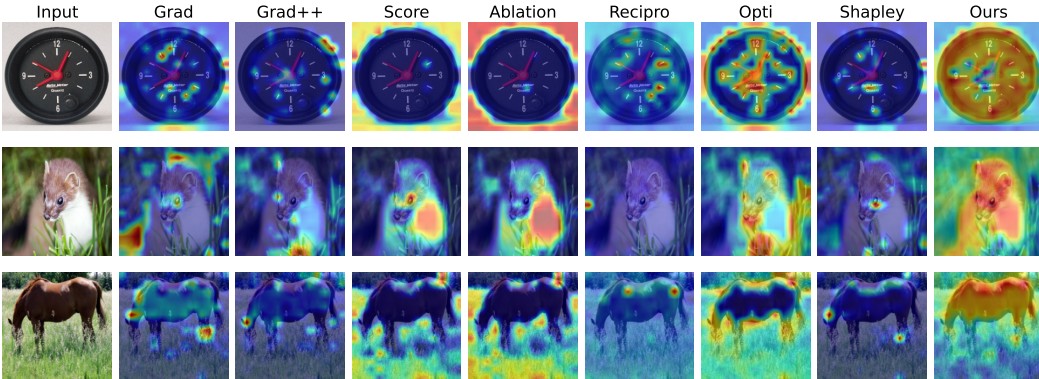

Figure 4: Visual explanations generated by various CAM-based methods for ViT-B.

## 4.4 ABLATION STUDY

To understand the impact of each component in our design, we conduct an ablation study by disabling or replacing modules related to clustering, discarding, and importance weighting. We compare the AD and IC of ClusCAM (full pipeline) to two groups of ablated variants. The first group replaces the clustering algorithm while keeping discarding and temperature-softmax: No clustering, spectral, and HDBSCAN clustering. The second group relies on K-Means++ but varies discarding and importance weighting: only logit, only softmax, discarding with softmax, and only temperature softmax. As shown in Fig. 5, all ablated variants underperform the full model, both in terms of AD and IC. This confirms that each component plays a complementary role in generating accurate and discriminative visual explanations.

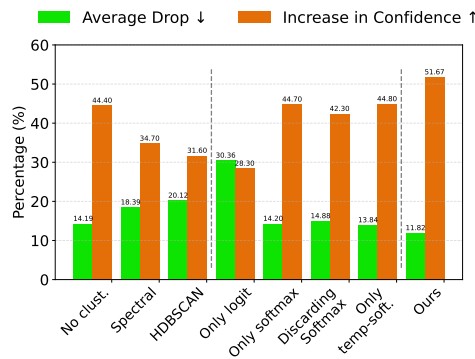

Figure 5: Ablation study results illustrating the contribution of three key components in our pipeline: clustering, discarding, and temperature softmax.

**Effect of clustering.** Removing clustering entirely or replacing it with baseline variants such as spectral clustering, or HDBSCAN consistently leads to lower IC scores (typically <45%) and moderately higher AD. This confirms that coherent groupings are crucial for constructing faithful explanations. More specifically, *Spectral clustering*, while theoretically powerful, reduces the feature space dimensionality, which often disrupts the spatial integrity necessary for accurate saliency (Von Luxburg, 2007). *HDBSCAN*, being density-based, tends to produce highly unbalanced or spatially fragmented clusters that fail to capture coherent regions of interest (Campello et al., 2013). By contrast, the *K-Means++* approach directly operates in the activation space, creating interpretable clusters where similar meta-representations are aggregated. Interestingly, the *No clustering* variant often outperforms

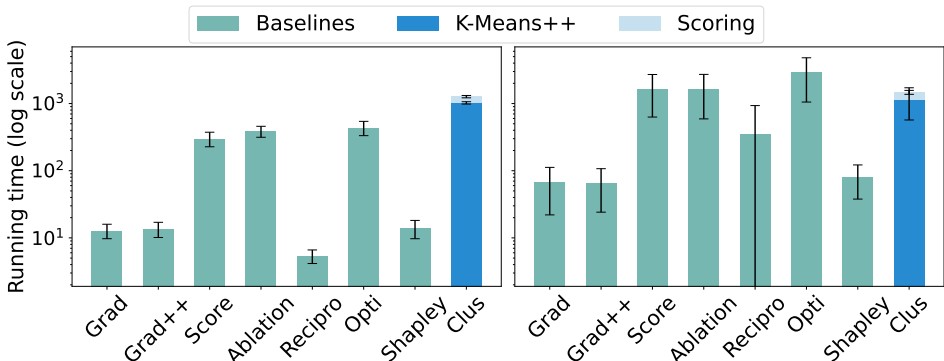

Figure 6: Average running times in the inference phase for different post-hoc explanation methods on CNNs (left) and ViTs (right). Here, ClusCAM is decomposed into K-Means and Scoring phases.

Spectral and HDBSCAN, though it still falls short of K-Means++. This implies that clustering is not universally beneficial; rather, the choice of a suitable clustering strategy is critical to effectively harness internal representations for faithful explanations. In fact, studies in self-supervised representation learning have shown that K-Means++ applied on feature embeddings can discover semantically meaningful clusters sufficient to drive representation learning without labels (Caron et al., 2018; 2020). Similarly, object discovery methods based on ViT rely on K-Means++ to group patch tokens into coherent foreground–background regions (Amir et al., 2021), further confirming that K-Means++ is a natural and effective choice for clustering deep features in explainability tasks.

**Effect of discarding and temperature softmax**. Disabling discarding or removing temperature scaling leads to clear performance degradation with up to 18.54% and 23.37% in AD and IC, respectively. This aligns with our intuition: without discarding, poorly relevant regions remain in the explanation. Without temperature scaling, the softmax weights become too uniform, reducing contrast between informative and uninformative regions. Notably, when using discarding without temperature scaling (*Discarding Softmax*), IC falls below 43%, indicating that raw softmax weighting cannot sufficiently emphasize high-scoring regions. Similarly, without discarding (*Only temp-soft.* and *Only softmax*) leads to lower precision, as noisy groups are retained. Moreover, temperature softmax shows a slight improvement compared to softmax. In summary, the best performance arises from the joint application of both modules: discarding and temperature softmax.

**Running time.** Regarding running time (Fig. 6), ClusCAM introduces a modest overhead from the K-Means++ initialization. However, this cost is offset by its efficient scoring phase, which requires only $K$ forward passes, in contrast to the hundreds needed by ScoreCAM or AblationCAM. As a result, the overall runtime of ClusCAM remains competitive. The advantage is even more pronounced on the ViTs, where the smaller number of patch tokens substantially reduces the cost of scoring-based methods compared to the CNNs.

Overall, the ablation confirms that clustering, discarding, and temperature softmax jointly contribute to faithful explanations with competitive runtime. Remaining issues include clustering overhead, heuristic hyperparameters, and evaluation limited to classification, which we discuss further in Appendix E.

## 5 CONCLUSIONS

We present ClusCAM, a novel gradient-free post-hoc explanation method that clusters internal representations into meta-representations and attributes their importance using discarding and temperature softmax mechanisms. Unlike conventional CAM-based methods that assess features independently and equally, ClusCAM accounts for high-level dependencies and interactions through group-wise attribution. Empirical results on both CNNs and ViTs demonstrate that ClusCAM consistently outperforms SoTA baselines across multiple quantitative metrics and produces more faithfully aligned explanations. These findings highlight that explicitly modeling inter-feature dependencies is essential for generating faithful and generalizable visual explanations in deep vision models for image classification tasks.

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

## THE USE OF LARGE LANGUAGE MODELS

LLMs were only used to improve the clarity and writing quality of the manuscript.

## A    IMPLEMENTATION DETAILS

All experiments were performed using an RTX 3090Ti GPU, with the code developed in Python version 3.12.2.

**Datasets.** For ImageNet (ILSVRC2012), we evaluated on 1,000 images from the validation set and 2,000 images for performance testing. In the ablation study, a reduced subset of 500 images was used due to resource constraints. For Alzheimer's disease classification, we employed an MRI dataset comprising four categories: Non-Demented, Very Mild Demented, Mild Demented, and Moderate Demented. Models were fine-tuned on a training set of 5,120 samples, validated on 380 images, and tested on 900 images. All images are resized to ($224 \times 224 \times 3$), scaled to the [0, 1] range, and normalized using a mean vector of [0.485, 0.456, 0.406] and a standard deviation vector of [0.229, 0.224, 0.225].

**Baselines.** We leveraged the codebase from the PyTorch-CAM library (Gildenblat & contributors, 2021), with the exception of ReciproCAM and OptiCAM, which were obtained from their respective GitHub repositories.

**ClusCAM.** The hyperparameters are detailed in Tab. 3. We also provide the complete code in the attached Supplementary Materials.

**Networks.** We utilized pre-trained networks, including CNNs such as the ResNet family (ResNet-18/34/50/101 (He et al., 2016)), EfficientNet (Tan & Le, 2019), and InceptionNet (Szegedy et al., 2016), as well as ViTs like ViT-B (Dosovitskiy et al., 2020), Swin-B (Liu et al., 2021), LeViT-129/256/'(Graham et al., 2021), CaiT-XXS-24 (Touvron et al., 2021), and PVTv2 (Wang et al., 2022) from the PyTorch model zoo (PyTorch Team). For CNN-based models, saliency maps were generated by hooking into the final convolutional layer, while for transformer-based models, we hooked immediately after either the patch embedding layer or the final convolutional layer.

Table 3: The hyperparameters used for ClusCAM implementation.

| Arch. | $K$ | $r(\%)$ | $\tau$ | Arch. | $K$ | $r(\%)$ | $\tau$ |
|---|---|---|---|---|---|---|---|
| ResNet-18 | 30 | 37.12 | 0.40 | ViT-B | 40 | 14.93 | 0.51 |
| ResNet-34 | 40 | 36.09 | 0.37 | Swin-B | 45 | 14.04 | 0.50 |
| ResNet-50 | 45 | 51.71 | 0.31 | LeViT-192 | 45 | 15.32 | 0.48 |
| ResNet-101 | 50 | 54.17 | 0.30 | LeViT-256 | 50 | 19.52 | 0.42 |
| EfficientNet | 50 | 40.78 | 0.33 | CaiT-XXS-24 | 40 | 16.02 | 0.50 |
| InceptionV3 | 45 | 45.08 | 0.33 | PVTv2 | 45 | 16.25 | 0.47 |

## B    SELECTING HYPERPARAMETERS

**Number of clusters.** Increasing $K$ improves semantic granularity but also introduces risks such as over-segmentation, increased computational cost, and reduced interpretability due to noisy or fragmented groups. As mentioned in section 3, we address this via a curvature-based Elbow strategy that captures the point of diminishing returns in a principled and data-driven manner.

Alg. 2 describes our strategy. First, we evaluate a performance proxy $P(K)$ across a range of candidate group sizes. This proxy quantifies the average gain in logit confidence when internal representations are partitioned into $K$ groups and used to generate masked inputs. To identify the "elbow" point, where increasing $K$ yields diminishing returns, we compute the discrete curvature $C(K_j)$ based on changes in $P(K)$ and the spacing between candidate values. The optimal group number $K^*$ is then chosen as the point with the maximum curvature, reflecting the most informative yet compact grouping.

---

**Algorithm 2** Optimal Group Number Selection via Normalized Elbow

---

**Input:** Validation set $V$, trained vision model $f$, target class $c$, candidate group sizes $\{K_1, \ldots, K_M\}$ in ascending order.
**Output:** Optimal number of groups $K^*$.
**Procedure:**

1: **for** $K$ in $\{K_1, \ldots, K_M\}$ **do**
2:     Compute proxy:
3:     $P(K) \leftarrow \frac{1}{K|V|} \sum_x \left( \sum_i \left( f_{\text{logit}}^c(x^{(i)}) - f_{\text{logit}}^c(x) \right) \right)$;
4: **end for**
5: **for** $j \leftarrow 3$ to $M$ **do**
6:     Compute proxy change:
7:       $\Delta P(K_j) \leftarrow P(K_j) - P(K_{j-1})$;
8:     Compute spacing:
9:       $\Delta K(K_j) \leftarrow K_j - K_{j-1}$;
10:     Compute normalized gain:
11:       $S(K_j) \leftarrow \Delta P(K_j)/\Delta K(K_j)$;
12:     Compute discrete curvature:
13:       $C(K_j) \leftarrow \left( S(K_j) - S(K_{j-1}) \right)/\Delta K(K_j)$;
14: **end for**
15: $K^* \leftarrow \arg\max_{j=3,\ldots,M} C(K_j)$;
16: **return** $K^*$

---

**Algorithm 3** Discarding Ratio Estimation via GMM Posterior Expectation

---

**Input:** Score matrix $\mathbf{S} \in \mathbb{R}^{N \times K}$ from validation set.
**Output:** Estimated discarding ratio $r \in (0, 1)$.
**Procedure:**

1: Flatten score matrix: $\mathcal{S} \leftarrow \texttt{Flatten}(\mathbf{S})$;
2: Fit 2-component Gaussian Mixture Model to $\mathcal{S}$;
3: Identify salient component:
4:     $\texttt{salient} \leftarrow \arg\max_{c \in \{1,2\}} \mu_c$;
5: Compute posterior probabilities:
6:     $\forall s \in \mathcal{S}, \quad p_{\text{non}}(s) \leftarrow \mathbb{P}(z = \text{non-salient} \mid s)$;
7: Estimate discarding ratio:
8:     $r \leftarrow \frac{1}{|\mathcal{S}|} \sum_{s \in \mathcal{S}} p_{\text{non}}(s)$;
9: **return** $r$

---

**Discarding ratio $r$.** We aim to determine the discarding ratio $r \in (0, 1)$, the fraction of groups to discard, using a probabilistic approach based on data.

Let $\mathbf{S} \in \mathbb{R}^{N \times K}$ be the matrix of group importance scores from a validation set of $N$ images, each with $K$ groups. We flatten this into a vector $\mathcal{S}$ and fit a two-component Gaussian Mixture Model (GMM) to model the score distribution:

$$p(s) = \pi_1 \cdot \mathcal{N}(s \mid \mu_1, \sigma_1^2) + \pi_2 \cdot \mathcal{N}(s \mid \mu_2, \sigma_2^2),$$

where $\pi_1, \pi_2$ are mixture weights and $\mu_c, \sigma_c^2$ are the mean and variance of each Gaussian component $c \in \{1, 2\}$. We assume one component captures salient groups and the other corresponds to non-salient (noise) groups.

We identify the non-salient component as the one with the lower mean, e.g., if $\mu_1 < \mu_2$, then component 1 is non-salient. For each score $s \in \mathcal{S}$, we compute the posterior probability of belonging to the non-salient class:

$$\mathbb{P}(z = \text{non-salient} \mid s) = \frac{\pi_{\text{non}} \cdot \mathcal{N}(s \mid \mu_{\text{non}}, \sigma_{\text{non}}^2)}{p(s)}.$$

The discarding ratio $r$ is then estimated as the expected proportion of non-salient scores. The full procedure is summarized in Alg. 3.

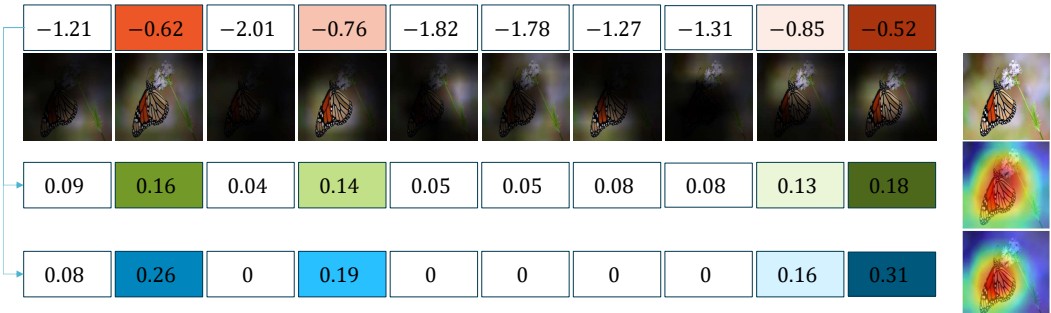

Figure 7: The change in importance scores using softmax (middle row) and temperature softmax with discarding (bottom row). The latter sharpens the saliency map. The higher the score, the more important the meta-representation.

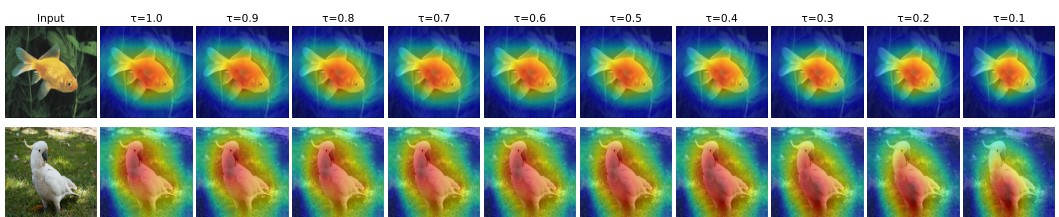

Figure 8: Effect of temperature $\tau$ on the quality of saliency maps on ResNet-18. As $\tau$ decreases from 1.0 to 0.1 (left to right), the highlighted regions become sharper and more localized. The maps show best visual clarity and semantic focus when $\tau$ is in the range $[0.3; 0.5]$.

**Temperature-scaled** $\tau$. We visualize the effect of temperature $\tau$ on the quality of saliency maps in Fig. 8. As $\tau$ decreases, the saliency maps become progressively more focused and concentrated, highlighting sharper and more localized regions. This reflects a stronger confidence in specific spatial activations. Conversely, when $\tau$ increases (e.g., $\tau \geq 0.9$), the maps become more diffuse and less discriminative, often highlighting large, ambiguous areas. Empirically, we observe that saliency maps generated with $\tau \in [0.3, 0.5]$ yield the best visual clarity and semantic relevance. Moreover, we show that temperature softmax with discarding can sharpen the saliency map in Fig. 7.

## C  EVALUATION METRICS

Given a model $f$ and the saliency map (explanation) $E^c$ generated from the test image $x$ with the class of interest $c$. Let $p = f(x)$ and $\tilde{p} = f(x \odot E^c)$. For localization ability, $B_p$ and $B$ are the predicted bounding box and the ground truth bounding box, respectively. Here, $B_p$ is generated by binarizing the saliency map by thresholding at its average value. Moreover, to be simple, we only consider the ground truth bounding box containing only one box, similar to the experiment in (Wang et al., 2020). We report five standard metrics used:

**(1) Average Drop (AD)** (Chattopadhay et al., 2018), lower is better, measures the drop in confidence when only the explanation region is shown:

$$\text{AD} := \frac{1}{N} \sum_{i=1}^{N} \frac{\max(0, p_i - \tilde{p}_i)}{p_i}. \tag{11}$$

**(2) Increase in Confidence (IC)** (Chattopadhay et al., 2018), higher is better, proportion of samples where model confidence increases after masking:

$$\text{IC} := \frac{1}{N} \sum_{i=1}^{N} \mathbf{1}[\tilde{p}_i > p_i] \tag{12}$$

**(3) Average Gain (AG)** (Zhang et al., 2024), higher is better, quantifies how much predictive power, measured as class probability, is gained when we mask the image:

$$\text{AG} := \frac{1}{N} \sum_{i=1}^{N} \frac{\max(0, \tilde{p}_i - p_i)}{1 - p_i} \tag{13}$$

**(4) Energy Pointing game (EP)** (Wang et al., 2020), higher is better, extracts the maximum point in the saliency map to see whether the maximum falls into the object bounding box:

$$\text{EP} := \frac{\sum_{(i,j) \in B} E^c(i,j)}{\sum_{(i,j)} E^c(i,j)}, \tag{14}$$

where $E^c(i,j)$ is the pixel at coordinates $(i,j)$ of $E^c$.

**(5) Localization Error (LE)** (Zhang et al., 2024), lower is better, measures the maximum overlap of the predicted bounding box with any ground truth bounding box:

$$\text{LE} := 1 - \text{IoU}(B, B_p), \tag{15}$$

where IoU is intersection over union.

## D    ADDITIONAL RESULTS

### D.1    INTERNAL REPRESENTATION COMBINATION

We empirically show that combining internal representations can increase the model confidence in Fig. 9. The logit change represents the model confidence; higher is better. Across all clusters, meta representations (red stars) consistently yield higher logit shifts than the internal cluster means (green triangles), indicating that the meta representations are more influential than the average behavior of the group. This suggests that our representation clustering mechanism effectively combines high-impact feature maps rather than simply using the internal representation independently. This supports the motivation behind ClusCAM's selection strategy, which prioritizes semantic saliency.

### D.2    OBJECT LOCALIZATION

Localization metrics evaluate how accurately saliency maps align with the ground truth bounding boxes of target foreground objects. While these metrics stem from the weakly supervised object localization task, their objectives differ from those of model explanation, as contextual information, often outside the object itself, can significantly influence a DNN's decision (Shetty et al., 2019; Rao et al., 2022). This misalignment is further reinforced by the findings of Zhang et al. (2024), who analyze the contributions of the object and its surrounding context to the model's decision. Their results show that using the ground truth bounding box alone, as a proxy saliency map, can degrade classification performance, even more so than its complement. Moreover, combining the bounding box with standard saliency maps often worsens performance across multiple metrics. These insights demonstrate that localization metrics, which rely solely on object-bound regions, fail to capture the full decision-making behavior of deep networks and are thus inadequate for evaluating interpretability methods. Nevertheless, we still report the results in Tab. 4. No single method consistently leads across all backbone architectures, except for ClusCAM. While ReciproCAM and ScoreCAM perform strongly on specific CNN models (e.g., ResNet-50, InceptionV3), ClusCAM demonstrates competitive localization performance on transformer-based backbones, achieving the lowest localization error.

### D.3    COMPARISON WITH VIT-SPECIFIC BASELINES

To further evaluate the effectiveness of ClusCAM on transformer-based architectures, we compare our method with Attention-Guided CAM (AG-CAM) (Leem & Seo, 2024), a recent explanation approach specifically designed for Vision Transformers. AG-CAM combines forward attention and backward gradient information to suppress noise and improve localization.

Table 5 reports the quantitative comparison on ViT-B using three standard explanation metrics.

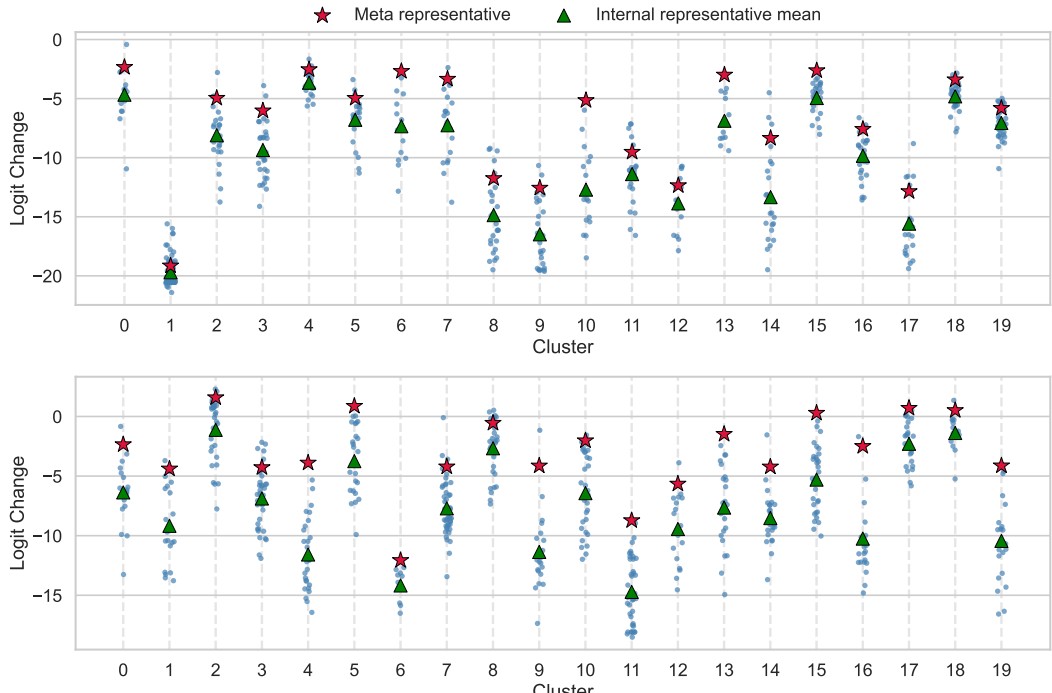

Figure 9: Comparison of individual feature map logit shifts with their corresponding cluster representations across two random samples. Each blue dot represents the logit shift of a feature map within a specific cluster. Red stars denote the logit shift of the meta representation, while green triangles indicate the mean of the internal representation's logit shift.

Table 4: Localization metrics for various CAM-based approaches across 12 different architectures on the ILSVRC dataset (Russakovsky et al., 2015). EP: Energy Pointing game; LE: Localization Error; ↓ / ↑: lower/higher is better. The best is highlighted in **bold**.

| Method | ResNet-18 | | ResNet-34 | | ResNet-50 | | ResNet-101 | | EfficientNet | | InceptionV3 | |
|---|---|---|---|---|---|---|---|---|---|---|---|---|
| | EP (↑) | LE (↓) | EP (↑) | LE (↓) | EP (↑) | LE (↓) | EP (↑) | LE (↓) | EP (↑) | LE (↓) | EP (↑) | LE (↓) |
| GradCAM | 51.60 | 74.33 | 51.81 | 73.91 | 53.34 | 73.18 | 53.35 | 73.13 | 52.27 | 82.15 | 55.44 | 71.06 |
| GradCAM++ | 51.49 | **72.89** | 51.73 | **72.89** | 53.21 | **71.68** | 53.30 | 71.78 | 53.14 | 83.71 | 55.19 | 70.14 |
| ScoreCAM | 50.99 | 73.55 | 50.82 | 73.94 | 52.64 | 72.29 | 52.32 | 73.58 | **53.49** | 88.18 | 53.65 | 73.01 |
| AblationCAM | 51.53 | 73.50 | **51.84** | 73.23 | 53.30 | 71.89 | 53.33 | 72.09 | 52.30 | 82.16 | 55.18 | 70.53 |
| ReciproCAM | **51.88** | 77.12 | 51.44 | 75.06 | **53.64** | 76.35 | **53.74** | 76.46 | 52.34 | 84.35 | **57.21** | 77.57 |
| OptiCAM | 48.76 | 75.31 | 48.10 | 75.84 | 50.09 | 74.11 | 52.17 | **68.68** | 51.80 | 79.97 | 54.19 | **69.53** |
| ShapleyCAM | 51.59 | 74.71 | 51.79 | 74.21 | 53.33 | 73.78 | 53.41 | 73.54 | 52.21 | 82.31 | 55.42 | 71.31 |
| ClusCAM | 50.45 | 73.66 | 50.10 | 74.62 | 52.28 | 72.98 | 51.94 | 73.29 | 51.09 | **72.29** | 53.12 | 74.11 |

| Method | ViT-B | | Swin-B | | LeViT-192 | | LeViT-256 | | CaiT-XXS-24 | | PVTv2 | |
|---|---|---|---|---|---|---|---|---|---|---|---|---|
| | EP (↑) | LE (↓) | EP (↑) | LE (↓) | EP (↑) | LE (↓) | EP (↑) | LE (↓) | EP (↑) | LE (↓) | EP (↑) | LE (↓) |
| GradCAM | 47.84 | 90.64 | 45.72 | 97.46 | 45.22 | 87.82 | 40.25 | 88.48 | 46.14 | 91.06 | 23.99 | 97.53 |
| GradCAM++ | 47.02 | 89.85 | 47.90 | 87.72 | 45.55 | 87.40 | 46.15 | 86.95 | 47.57 | 87.01 | 16.92 | 98.14 |
| ScoreCAM | 47.05 | 86.31 | 49.30 | 84.40 | 48.39 | 84.94 | 49.86 | 82.99 | 46.47 | 90.57 | 10.01 | 98.75 |
| AblationCAM | 46.24 | 85.24 | **49.66** | 82.40 | 44.30 | 88.38 | 32.13 | 87.32 | 46.67 | 93.26 | 25.57 | 96.72 |
| ReciproCAM | 48.08 | 85.66 | 47.11 | 80.35 | 48.50 | 85.12 | 49.01 | 84.81 | 46.84 | 71.82 | 50.06 | 80.76 |
| OptiCAM | **48.56** | 79.32 | 49.02 | 81.43 | 48.58 | 74.20 | **49.91** | 72.83 | **49.10** | 78.39 | 51.19 | 73.06 |
| ShapleyCAM | 47.98 | 91.00 | 46.16 | 97.27 | 42.70 | 85.89 | 41.76 | 84.56 | 43.99 | 89.99 | 10.93 | 99.09 |
| ClusCAM | 46.75 | **71.59** | 46.96 | **68.70** | 47.47 | **71.37** | 47.77 | **70.24** | 47.27 | **68.83** | 46.44 | **69.39** |

As shown in Table 5, AG-CAM performs competitively among existing CAM-based approaches and clearly outperforms several methods originally developed for CNN architectures, such as GradCAM, GradCAM++, and ScoreCAM. This confirms its effectiveness as a ViT-specific explanation baseline.

However, compared with approaches explicitly adapted for or generalized to transformer-based models, such as OptiCAM and our ClusCAM, AG-CAM achieves lower performance on all three metrics,

Table 5: Quantitative comparison of explanation methods on ViT-B.

| Method | AD ↓ | IC ↑ | AG ↑ |
|---|---|---|---|
| GradCAM | 73.40 | 7.20 | 4.54 |
| GradCAM++ | 74.46 | 7.45 | 5.32 |
| ScoreCAM | 56.03 | 19.10 | 15.50 |
| AblationCAM | 57.94 | 15.25 | 20.14 |
| ReciproCAM | 53.46 | 9.00 | 8.68 |
| OptiCAM | **4.69** | 36.30 | 13.12 |
| ShapleyCAM | 74.44 | 5.21 | 1.22 |
| ClusCAM (ours) | 5.21 | **60.75** | **73.96** |
| AG-CAM | 17.61 | 22.35 | 6.28 |

particularly on IC and AG. Notably, while AG-CAM is tailored to the original ViT architecture, ClusCAM is designed as an architecture-agnostic framework and can be applied consistently to both CNNs and a wide range of ViT variants, including Swin, LeViT, CaiT, and PVT. This highlights the stronger generalization capability of ClusCAM across different transformer designs.

### D.4 Detail results on ILSVRC and Alzheimer's datasets

We report the detailed quantitative results of different CAM-based approaches on the ILSVRC and Alzheimer's datasets in Tab. 6 and Tab. 7, respectively. These results allow a comprehensive comparison across both CNN and VT backbones, providing insights into the effectiveness and generalizability of ClusCAM under different architectures and domains.

### D.5 Trade-off between faithfulness and runtime

ClusCAM is designed as a faithful gradient-free attribution method, rather than a real-time explanation system for latency-critical applications. As is common in gradient-free explainability approaches (Wang et al., 2020; Ramaswamy et al., 2020), it trades additional computation for improved attribution quality. This section analyzes this trade-off in terms of performance gains, runtime overhead, and computational complexity.

**Attribution performance vs. runtime**. Table 8 reports the relative gains in attribution performance and runtime compared to GradCAM, for the two strongest gradient-free baselines (ScoreCAM and OptiCAM) and our ClusCAM, evaluated on both CNNs and ViTs. All values are reported as multiplicative factors with respect to GradCAM.

On CNNs, ClusCAM exhibits a higher runtime overhead due to the additional clustering step, resulting in a $\times 44.24$ slowdown compared to GradCAM. However, this overhead is proportional to the significant gains in attribution quality, achieving the best improvements across all three metrics (AD, IC, AG). Importantly, its runtime remains within the same order of magnitude as other gradient-free methods such as ScoreCAM and OptiCAM.

Interestingly, this trade-off becomes more favorable on Vision Transformers. While still improving attribution performance substantially over both baselines, ClusCAM achieves the lowest runtime among the three gradient-free methods on ViTs. This behavior arises from the reduced number of internal representations in ViTs compared to CNNs.

**Efficiency in terms of forward passes**. One of the main sources of computational overhead in gradient-free methods lies in repeated forward passes (FP). Table 9 compares the number of required forward passes for several representative explanation methods.

Unlike ScoreCAM and AblationCAM, whose computational cost scales linearly with the number of internal representations $N$, ClusCAM requires only $K$ forward passes, where $K \ll N$ (e.g., for ResNet18, $K = 30$ while $N = 512$). This significantly reduces the cost of the scoring phase, making ClusCAM competitive among gradient-free CAM methods.

Table 6: Evaluation of various CAM-based approaches across 12 different architectures on the ILSVRC dataset (Russakovsky et al., 2015). AD: Average Drop; IC: Increase in Confidence; AG: Average Gain; ↓ / ↑: lower/higher is better. The best is highlighted in **bold** while the second rank is in *italics*.

| Method | ResNet-18 | | | ResNet-34 | | | ResNet-50 | | | ResNet-101 | | |
|---|---|---|---|---|---|---|---|---|---|---|---|---|
| | AD (↓) | IC (↑) | AG (↑) | AD (↓) | IC (↑) | AG (↑) | AD (↓) | IC (↑) | AG (↑) | AD (↓) | IC (↑) | AG (↑) |
| GradCAM | 21.36 | 32.15 | 13.24 | 17.80 | 35.35 | 16.53 | 14.62 | 38.70 | 19.52 | 13.50 | 42.20 | 22.28 |
| GradCAM++ | 22.09 | 29.80 | 11.75 | 18.22 | 34.20 | 15.22 | 14.87 | 38.00 | 18.10 | 13.45 | 40.75 | 20.99 |
| ScoreCAM | 15.90 | 41.20 | *18.19* | 11.94 | *50.20* | *26.25* | 9.74 | *53.75* | *28.96* | 8.42 | *57.60* | *34.83* |
| AblationCAM | 21.38 | 30.80 | 12.44 | 18.05 | 34.25 | 15.67 | 14.59 | 38.60 | 18.49 | 13.48 | 41.40 | 21.22 |
| ReciproCAM | 25.73 | 27.60 | 11.61 | 20.11 | 32.80 | 15.73 | 18.08 | 34.40 | 17.52 | 16.63 | 37.80 | 20.38 |
| OptiCAM | *11.96* | *42.10* | 14.21 | *10.76* | 41.90 | 14.58 | **7.35** | 45.5 | 15.14 | *7.57* | 43.75 | 15.64 |
| ShapleyCAM | 21.01 | 33.20 | 13.98 | 17.20 | 36.30 | 17.34 | 14.38 | 39.75 | 20.26 | 13.11 | 42.90 | 22.98 |
| ClusCAM (Ours) | **11.50** | **50.50** | **22.99** | **9.16** | **57.50** | **32.06** | *8.58* | **57.20** | **32.19** | **6.57** | **61.50** | **36.37** |

| Method | EfficientNet | | | InceptionV3 | | | ViT-B | | | Swin-B | | |
|---|---|---|---|---|---|---|---|---|---|---|---|---|
| | AD (↓) | IC (↑) | AG (↑) | AD (↓) | IC (↑) | AG (↑) | AD (↓) | IC (↑) | AG (↑) | AD (↓) | IC (↑) | AG (↑) |
| GradCAM | 26.72 | 30.35 | 11.28 | 19.57 | 31.65 | 20.64 | 73.40 | 7.20 | 4.54 | 95.63 | 0.50 | 0.57 |
| GradCAM++ | 31.52 | 25.30 | 9.27 | 19.85 | 32.05 | 19.91 | 74.46 | 7.45 | 5.32 | 68.72 | 1.90 | 1.55 |
| ScoreCAM | 32.73 | 31.35 | *13.65* | 9.24 | *53.85* | *38.31* | 56.03 | 19.10 | *15.50* | 47.70 | 8.10 | 9.93 |
| AblationCAM | 26.86 | 30.60 | 7.93 | 19.57 | 31.65 | 9.49 | 57.94 | 15.25 | 20.14 | 43.24 | 8.50 | 11.23 |
| ReciproCAM | 32.89 | 26.75 | 9.95 | 28.09 | 26.35 | 17.63 | 52.46 | 9.00 | 8.68 | 48.33 | 2.30 | 7.53 |
| OptiCAM | *6.93* | *35.80* | 4.99 | *7.93* | 41.95 | 19.74 | **4.69** | *36.30* | 13.12 | **7.20** | *21.60* | *21.95* |
| ShapleyCAM | 26.41 | 31.10 | 11.59 | 19.41 | 20.90 | 5.40 | 74.44 | 5.21 | 1.22 | 95.66 | 0.35 | 0.61 |
| ClusCAM (Ours) | **4.84** | **67.15** | **28.36** | **6.29** | **63.60** | **46.64** | *5.21* | **60.75** | **73.96** | *8.22* | **22.90** | **45.49** |

| Method | LeViT-192 | | | LeViT-256 | | | CaiT-XXS-24 | | | PVTv2 | | |
|---|---|---|---|---|---|---|---|---|---|---|---|---|
| | AD (↓) | IC (↑) | AG (↑) | AD (↓) | IC (↑) | AG (↑) | AD (↓) | IC (↑) | AG (↑) | AD (↓) | IC (↑) | AG (↑) |
| GradCAM | 62.16 | 8.50 | 1.54 | 64.58 | 6.15 | 1.06 | 72.26 | 3.45 | 0.88 | 92.02 | 1.30 | 1.07 |
| GradCAM++ | 61.99 | 8.60 | 1.44 | 59.16 | 7.05 | 1.15 | 64.09 | 3.80 | 1.02 | 94.76 | 0.90 | 0.38 |
| ScoreCAM | 33.21 | 28.90 | 8.19 | 28.09 | 27.60 | *6.32* | 72.49 | 4.85 | 1.64 | 99.71 | 0 | 0 |
| AblationCAM | 63.00 | 8.45 | 1.43 | 55.19 | 11.40 | 1.39 | 83.58 | 1.80 | 0.41 | 88.21 | 2.15 | 1.03 |
| ReciproCAM | 36.29 | 15.85 | 3.10 | 28.73 | 14.70 | 3.12 | 23.36 | 10.30 | 2.20 | 53.62 | 10.20 | 4.14 |
| OptiCAM | *3.30* | *47.20* | *3.42* | *2.35* | *50.85* | 2.84 | **3.02** | *43.05* | *7.50* | **4.15** | *48.7* | *5.20* |
| ShapleyCAM | 59.88 | 9.70 | 1.93 | 52.85 | 10.80 | 1.42 | 67.60 | 3.45 | 1.05 | 97.03 | 0.70 | 0.56 |
| ClusCAM (Ours) | **1.33** | **80.25** | **20.30** | **1.55** | **74.75** | **15.52** | *5.51* | **41.80** | **15.38** | *12.03* | **47.80** | **16.66** |

Moreover, the clustering step using K-means++ is more efficient for Vision Transformers, as ViTs typically have fewer internal representations (e.g., 196 patch tokens in ViT-B versus 512 feature maps in ResNet18). As a result, the overall runtime overhead of ClusCAM is notably reduced on ViTs.

Together, these analyses demonstrate that ClusCAM achieves a well-balanced trade-off between computational cost and attribution accuracy, especially for transformer-based architectures.

## E  DISCUSSION

Our study highlights the importance of modeling interactions between internal representations when generating saliency maps. By clustering activations into meta-representations, ClusCAM captures compositional structures that traditional CAM variants often overlook. This group-wise attribution leads to sharper and more faithful explanations. The discarding mechanism and temperature scaling further refine the final explanations by suppressing spurious groups and emphasizing the most relevant ones. Notably, our method generalizes effectively across both CNNs and ViTs, outperforming existing methods on a wide range of architectures and metrics.

Nonetheless, while the proposed method shows strong empirical performance, several limitations remain. First, ClusCAM introduces additional computational overhead compared to conventional CAM variants. The clustering of internal representations increases inference time, especially on large-scale models. Although faster than exhaustive methods like ScoreCAM, AblationCAM, and OptiCAM on ViTs, a promising direction for improvement is to design more efficient clustering

Table 7: Evaluation of various CAM-based approaches across 12 different architectures on the Alzheimer's dataset (Falah.G.Salieh, 2023). AD: Average Drop; IC: Increase in Confidence; AG: Average Gain; ↓ / ↑: lower/higher is better. The best is highlighted in **bold** while the second rank is in *italics*.

| Method | ResNet-18 AD (↓) | IC (↑) | AG (↑) | ResNet-34 AD (↓) | IC (↑) | AG (↑) | ResNet-50 AD (↓) | IC (↑) | AG (↑) | ResNet-101 AD (↓) | IC (↑) | AG (↑) |
|---|---|---|---|---|---|---|---|---|---|---|---|---|
| GradCAM | 0.45 | 25.49 | 16.98 | 1.73 | 60.13 | 42.49 | 8.66 | 2.97 | 1.18 | 21.11 | 19.70 | 11.40 |
| GradCAM++ | 0.43 | 28.85 | 20.30 | 1.83 | 58.87 | 42.66 | 8.24 | 3.44 | 1.34 | 20.97 | 19.62 | 12.05 |
| ScoreCAM | 0.38 | 34.01 | 24.36 | 1.06 | *79.52* | 68.98 | 3.30 | 13.21 | 7.62 | 13.49 | 32.40 | 22.96 |
| AblationCAM | 0.42 | 28.46 | 19.56 | 1.70 | 62.00 | 45.40 | 3.38 | 9.85 | 5.56 | 17.17 | 22.52 | 14.76 |
| ReciproCAM | 0.48 | 22.91 | 15.32 | 1.74 | 61.69 | 42.11 | 9.98 | 2.66 | 1.02 | 21.72 | 17.90 | 10.19 |
| OptiCAM | *0.31* | **58.72** | *25.89* | **0.62** | 73.03 | 20.37 | **0.03** | 38.31 | 26.50 | **1.23** | 37.29 | *26.71* |
| ShapleyCAM | 0.44 | 25.57 | 16.89 | 1.73 | 60.67 | 43.17 | 8.70 | 2.89 | 1.09 | 21.25 | 18.06 | 9.65 |
| ClusCAM (Ours) | **0.25** | *47.30* | **35.21** | *0.98* | **84.28** | **75.07** | *0.52* | **79.51** | **42.28** | *9.98* | **76.08** | **52.26** |

| Method | EfficientNet AD (↓) | IC (↑) | AG (↑) | InceptionV3 AD (↓) | IC (↑) | AG (↑) | ViT-B AD (↓) | IC (↑) | AG (↑) | Swin-B AD (↓) | IC (↑) | AG (↑) |
|---|---|---|---|---|---|---|---|---|---|---|---|---|
| GradCAM | 54.32 | 19.39 | 66.18 | 21.27 | 64.43 | 57.07 | 7.43 | 57.86 | 44.54 | 81.61 | 3.44 | 2.43 |
| GradCAM++ | 50.34 | *23.46* | 70.11 | 20.93 | 64.11 | 56.10 | 7.43 | 52.54 | 40.33 | 55.70 | 6.33 | 3.98 |
| ScoreCAM | *45.70* | 21.03 | 67.02 | 19.26 | *71.62* | 65.30 | 7.33 | 69.27 | 53.30 | 57.03 | 7.43 | 5.19 |
| AblationCAM | 54.34 | 19.62 | 66.39 | 21.05 | 64.43 | 56.93 | 7.43 | 55.90 | 42.39 | 74.86 | 0.47 | 0.19 |
| ReciproCAM | 51.33 | 20.95 | 67.70 | 21.00 | 63.96 | 56.49 | 7.43 | 49.49 | 37.89 | 78.86 | 1.64 | 0.80 |
| OptiCAM | 49.09 | 22.44 | *69.27* | **5.76** | 67.79 | 35.43 | **2.73** | 58.64 | 19.22 | **3.71** | **45.27** | **29.45** |
| ShapleyCAM | 55.80 | 17.90 | 64.80 | 21.13 | 64.43 | 57.10 | 7.40 | 55.36 | 43.14 | 81.38 | 3.67 | 2.52 |
| ClusCAM (Ours) | **38.38** | **28.77** | **75.47** | *17.39* | **74.04** | **69.00** | *6.10* | **76.39** | **57.10** | *4.86* | *37.29* | *25.55* |

| Method | LeViT-192 AD (↓) | IC (↑) | AG (↑) | LeViT-256 AD (↓) | IC (↑) | AG (↑) | CaiT-XXS-24 AD (↓) | IC (↑) | AG (↑) | PVTv2 AD (↓) | IC (↑) | AG (↑) |
|---|---|---|---|---|---|---|---|---|---|---|---|---|
| GradCAM | 88.13 | 2.89 | 0.69 | 21.21 | 8.37 | 0.50 | 64.68 | 7.58 | 1.27 | 34.45 | 18.30 | 3.41 |
| GradCAM++ | 85.00 | 3.67 | 0.86 | 21.53 | 10.95 | 0.77 | 54.74 | 6.80 | 1.06 | 48.57 | 8.91 | 2.48 |
| ScoreCAM | 46.20 | 32.13 | 4.60 | 18.85 | 14.39 | 0.71 | 54.04 | 11.42 | 2.18 | 52.73 | 8.21 | 2.47 |
| AblationCAM | 72.35 | 4.69 | 0.86 | 18.52 | 1.88 | 0.46 | 84.31 | 0.78 | 0.17 | 28.54 | 20.41 | 3.74 |
| ReciproCAM | 99.49 | 0.08 | 0.01 | 18.34 | 20.80 | 0.83 | 32.32 | 13.06 | 1.71 | 6.85 | 48.63 | *5.34* |
| OptiCAM | **14.71** | *28.77* | *3.20* | 29.69 | 11.10 | 0.59 | **0.89** | *74.90* | *3.88* | **1.84** | *59.11* | 3.39 |
| ShapleyCAM | 84.52 | 4.85 | 1.05 | 20.10 | 9.07 | 0.45 | 56.18 | 7.97 | 1.10 | 34.84 | 17.04 | 3.35 |
| ClusCAM (Ours) | *25.19* | **41.91** | **6.06** | *12.60* | **29.24** | **1.11** | *0.94* | **78.89** | **6.68** | *3.04* | **67.71** | **8.15** |

Table 8: Trade-off analysis: multiplicative gains in performance and runtime compared to GradCAM.

| Architecture | Metric | ScoreCAM | OptiCAM | ClusCAM |
|---|---|---|---|---|
| CNNs | AD | ×1.29 | ×2.16 | ×2.42 |
| | IC | ×1.37 | ×1.19 | ×1.70 |
| | AG | ×1.52 | ×0.81 | ×1.92 |
| | Runtime | ×23.43 | ×34.11 | ×44.24 |
| ViTs | AD | ×1.36 | ×18.61 | ×13.60 |
| | IC | ×3.27 | ×9.13 | ×12.10 |
| | AG | ×4.30 | ×5.59 | ×19.39 |
| | Runtime | ×24.90 | ×43.69 | ×22.16 |

algorithms that can retain grouping power while reducing the computational burden, since the scoring phase itself already incurs negligible cost.

Second, the selection of hyperparameters $(K, r, \tau)$, while guided by principles such as curvature-based saturation (for $K$), Gaussian mixture modeling (for $r$), and temperature scaling heuristics (for $\tau$), currently lacks a strong theoretical foundation. Although our ablation study confirms their empirical effectiveness, future work could aim to derive stronger theoretical guarantees or formulate principled optimization objectives that justify these design choices.

Table 9: Number of forward passes (FP) required by different explanation methods ($K \ll N$).

| Method | GradCAM | GradCAM++ | ScoreCAM | AblationCAM | ClusCAM |
|---|---|---|---|---|---|
| Number of FP | 1 | 1 | $N$ | $N$ | $K$ |

Third, ClusCAM is currently evaluated only on image classification tasks. Its design, however, is not inherently limited to classification. Extending our method to dense prediction tasks such as semantic segmentation, object detection, or even video-based activity recognition could unlock its full potential. These tasks may require adapting the clustering mechanism to account for spatial continuity or temporal consistency, but the core idea of meta-representation attribution remains applicable. Addressing these limitations could improve both the scalability and generality of ClusCAM in real-world deployments.

As future work, we plan to extend ClusCAM towards a concept-based explanation framework, where each meta-representation is associated with a higher-level, human-interpretable visual concept rather than only a spatial attribution. In addition, a deeper investigation of the behavioral differences between CNNs and Vision Transformers under the proposed clustering and discarding mechanism remains an interesting direction, as their distinct inductive biases and representation structures may lead to different explanation dynamics and failure modes.

