# OpenReview forum: "ClusCAM: Clustered Visual Explanations for Vision Models in Image Classification"
_ICLR.cc/2026/Conference — Submitted to ICLR 2026_

### Official Review · Reviewer_rjyS · 2025-10-16

**Soundness:** 3
**Presentation:** 2
**Contribution:** 2
**Rating:** 4
**Confidence:** 4

**Summary:**

The paper is aiming at enhancing the XAI of CNNs and ViTs. The idea is to cluster the representations into K groups, then upsample and normalize them to be the same size as the input, and use them as a soft mask of the input. Then aggregate the difference between the logit activation of the masked input and the benign one. This approach is treating in a weighted manner the different concepts of the input. It is like projecting the input into concepts and highlight mostly the dominant ones.

**Strengths:**

* XAI is a very important topic, especially nowadays when it is crucial to better understand the inner process of networks.
* The Related Work section is detailed and informative, and the academic gap is well explained.
* The idea of combining concept fetching with explainability is intriguing.

**Weaknesses:**

* The approach is written as if it can be applied for both CNNs and ViTs. One of the gaps mentioned is that current approaches is overlooking weighting when aggregating representations. While it might hold for CNNs it is not true in ViTs, see for example [1], [2] and [3]. Moreover there is a large amount of dedicated approaches for XAI specifically on ViT which is failed to be mentioned.
* The method is primarily empirical with lack of intuitive explanation on the considerations for why applying each step behind it. For example why specifically using Kmeans++ (and not any other clustering scheme?)? why the element-wise product between the meta-representation to the input represents? Why is formula 5 is reasonable? what if it would be division? why removing the r% least important meta representations? (If they are not important for the classification, then they do not have impact anyway). In general it might be OK when there is empirical progression, however in my opinion it should come along with intuitive explanation so that follow-up researcher could extend your ideas. In my opinion it is lack in the current submission.
* Most of the enhancement is of very empirical steps like filtering out less relevant projections. I believe the authors need to clearly separate the conceptual novelty from the empirical contributions and to focus more on the conceptual contribution. Most of the paper elaborate on the empirical steps, which is with less academic improvement.

Minor weaknesses:
* confusing notations. H and W represent the input dimensions while h and w represent the representation dimensions. I would select other notations to make it clearer.
* The placement of visualizations is a bit awkward. For example, the algorithm is presented before the algorithm itself is explained.
* The initials ViT is more common for Vision Transformers than VT.
* r is used both for the ratio of the dropout and for the percentage of meta-representations to filter.
* What is the meaning of the colors in Fig. 2? If it is just to indicate different meta-representations, then it is not so clear.

refs:

[1] Transformer interpretability beyond attention visualization, Chefer et al. CVPR 21.

[2] Token transformation matters: Towards faithful post-hoc explanation for vision transformer. We et al. CVPR 24.

[3] From Attention to Prediction Maps: Per-Class Gradient-Free Transformer Explanations. Schaffer et al. PrePrint

**Questions:**

* It is known that there are polysemantic neurons. i.e., neurons which activated differently when facing different inputs [1]. How do you think your approach will be affected from this? specifically Im curious on the hard clustering step which i assume that will cause hard selection for a certain "meaning".
* The clustering stage is closely related to concept clustering which typically implemented through Sparse AutoEncoders (SAE). Have you tried implement it with SAEs ([2] for example but there are a lot of papers in this topic)? If so, then what are the results? If no, I would recommend try it since it is always better to lean on a grounded approaches.
* How does the normalization is done? Which method of Upsampling is applied?
* Why is M_j represents importance? The element-wise multiplication explained as a sort of soft masking when M is the soft mask matrix. Implicitly it is referred that the magnitude of the representation represents importance (personally I agree with this observation), but in your view, why do you think it holds?
* Why do you selected using dropout to filter out outliers? It is statistical operator, it can in some cases not filter them at all? Moreover, it is not a good practice that the inference is random (depend on the seed). It was found that "registers" might be the cause for outlier heads in ViT [3], at least for ViT it can be the starting point to find this outliers systematically instead of filter them statistically.


In general, I think that the paper is too much empirical in his nature, and the authors should clearly distill the pure conceptual contribution from the empirical steps. In some cases it is even better to get a method which is a bit less good in performance but much more understandable. In the case of your approach, I think that the approach is too empirical such that it is very hard to isolate the pure contribution. Moreover, I think that the authors should focus more on explaining the intuitions behind the approach steps, and to make it clearer what information is better seized using your approach.

refs:

[1] Interpreting the Second-Order Effects of Neurons in CLIP. Gandelsman et al. ICLR 25.

[2] Interpreting CLIP with Hierarchical Sparse Autoencoders. Zaigrajew et al. ICML 25.

[3] Vision Transformers Need Registers. Darcet et al. ICLR 24.

---

> ### Author Response · Authors · 2025-11-21
>
> Dear Reviewer rjyS,
>
> Thank you for spending the time to review our work and giving us constructive feedback. We clarify your concerns below:
> ___
> **[W1. Representation aggregation]**
>
> **For representation aggregation:** Our statement **was not intended to deny that ViTs already contain aggregation-weighting** mechanisms. Rather, our claim concerns a different form of weighting: existing CAM-style post-hoc attribution methods, although applicable to both CNNs and ViTs, treat each patch token as an independent unit of evidence at the explanation stage, which can lead to less faithful explanation maps. It is this missing **group-level attribution weighting** that ClusCAM addresses.
>
> **For mention ViT-specific XAI approaches:** we acknowledge this and will add a short paragraph in the Related Work section, summarizing major ViT-specific XAI approaches and how they differ from our setting.
>
> **For completeness on the CAM side**, we will include AG-CAM [1] as a ViT-specific CAM baseline (following the suggestion from the Reviewer ztRk). We report the results below:
>
> *Table 1. Quantitative comparison of explanation methods on ViT-B.*
> | Method       | AD ↓  | IC ↑  | AG ↑  |
> |-------------|-------|-------|-------|
> | GradCAM     | 73.40 | 7.20  | 4.54  |
> | GradCAM++   | 74.46 | 7.45  | 5.32  |
> | ScoreCAM    | 56.03 | 19.10 | 15.50 |
> | AblationCAM | 57.94 | 15.25 | *20.14* |
> | ReciprocCAM | 53.46 | 9.00  | 8.68  |
> | OptiCAM     | **4.69**  | *36.30* | 13.12 |
> | ShapleyCAM  | 74.44 | 5.21  | 1.22  |
> | ClusCAM     | *5.21* | **60.75** | **73.96** |
> | AG-CAM      | 17.61 | 22.35 | 6.28  |
>
> Note that AG-CAM is specifically designed for ViT architectures, while our method works well with both ViT and its variants. The results show that AG-CAM outperforms most methods that were originally developed for CNN-based models. However, when compared with approaches explicitly optimized for transformer backbones, such as OptiCAM and ClusCAM, AG-CAM performs less favorably. The updated results show that ClusCAM remains competitive and further strengthens our work.
> ___
> **[W2. Intuitive explanation for each step]**
>
> Each design choice in ClusCAM is grounded in established principles with a brief intuitive explanation. We clarify below:
>
> **(1) Why K-Means++?**
>
> As noted in Section 3.1 (lines 205 to 207), we use K-Means++ because spatially co-activated representations tend to cluster well under Euclidean distance, and the plus-plus initialization improves stability and convergence. Also, our ablation study (lines 424 to 453) empirically shows why K-Means++ yields meaningful spatial groups in ClusCAM, whereas alternative clustering methods (e.g., spectral clustering, HDBSCAN) performed less favorably. This is consistent with prior deep clustering and self-supervised vision work [2, 3], where Euclidean-feature clustering is effective for grouping deep representations.
>
> **(2) Why element-wise product (mask × input)?**
>
> This is the standard mechanism in CAM-based perturbation [4, 5, 6]. It retains only the regions emphasized by the meta-representation and ensures that the importance score reflects the contribution of those regions to the model prediction. We will revise the manuscript to make these points clearer.
>
> **(3) Why logit-difference in Eq. 5?**
>
> The logit-difference formulation is both theoretically meaningful and empirically stable. Measuring the difference between the masked-logit and the original logit is a canonical choice widely used in CAM literature [4, 5, 6] because it directly reflects how much the selected region contributes to the class evidence.
>
> On the other hand, the idea of using division instead of difference is problematic because it amplifies importance scores in an uncontrolled manner, which occurs when explaining a class different from the model’s predicted class (i.e., the logit of that class may be 0 or extremely close to 0).
>
> **(4) Why removing the r% least important groups?**
>
> We have briefly explained the motivation for removing r% groups in Sec. 3.3 (lines 227 to 232). Moreover, the “not important” groups introduce noise (see Fig. 7 in the appendix), and removing them increases attribution contrast and faithfulness. In addition, our ablation study shows that discarding a small fraction of the lowest-scoring groups improves AD and IC by 2.02% and 6.87%, respectively. That said, we acknowledge that providing a more concrete explanation would improve the intuitiveness of this design choice in the revised manuscript.
>
> In summary, we agree that the paper would benefit from making the intuitions behind ClusCAM’s design more explicit in the revised version.

---

> > ### Author Response · Authors · 2025-11-21
> >
> > **[W.3 Conceptual vs. empirical contributions]**
> >
> > We would like to clarify the distinction between the conceptual and empirical contributions of ClusCAM:
> >
> > * **Conceptual contribution.** ClusCAM introduces group-wise attribution as a **new** explanatory principle.
> > Rather than treating individual feature maps or tokens as isolated units, we observe that deep networks encode visual concepts through co-activated groups of internal features. ClusCAM formalizes this idea by (i) constructing meta-representations based on co-activation similarity, (ii) scoring each group by its functional contribution through logit-difference perturbation, and (iii) aggregating group-wise contributions to form the final saliency map. This conceptual shift from feature-wise to group-wise attribution is detailed in Section 3.1 and formalized in Algorithm 1, and its validity is empirically supported in Section 4.1.
> >
> > * **Empirical contribution.** The empirical components of ClusCAM include K-Means++ grouping, deterministic dropout, and temperature-scaled weighting, whose combined effectiveness is evaluated across 12 CNN- and ViT-based models. Each is an operational instantiation of the group-wise attribution framework:
> >
> >     * Clustering defines the meta-representation space on which group-wise reasoning is carried out.
> >
> >     * Dropout removes spurious groups that would otherwise distort group-level evidence.
> >
> >     * Temperature scaling modulates the sharpness of the group contribution distribution.
> >
> > Our ablation study (Section 4.4) shows how each empirical component supports the conceptual framework by improving stability, suppressing noise, and enhancing attribution quality. These steps refine and reinforce the underlying conceptual idea rather than introducing unrelated empirical tricks. We will revise the paper to highlight this conceptual and empirical structure more clearly.
> > ___
> > **[Minor weaknesses in notation]**
> >
> > We thank the reviewer for pointing out this, and we will revise the notation to avoid ambiguity (e.g., use distinct symbols for input and representation dimensions), adopt the standard “ViT” terminology, and reorganize the visualization and algorithm presentation for better clarity and flow.
> > ___
> >
> > **[Q1. Polysemantic neurons & effect on hard clustering]**
> >
> > Polysemantic neurons are **unlikely to be an issue** in ClusCAM because they indeed encode multiple concepts depending on the input context. More importantly, ClusCAM does not intend to assign semantic meaning to individual feature maps, but instead to group feature maps (or tokens) that produce similar activation patterns on the current input.
> > In other words, clustering is input-specific, not global, and the meta-representations capture the co-activation structure for a particular image. Therefore, even if a neuron is polysemantic across the dataset, its activation for one input typically lies in a well-defined region of activation space, and grouping helps capture such co-activated patterns.
> > ___
> > **[Q2. Clustering vs. Sparse Autoencoders]**
> >
> > Sparse Autoencoders (SAEs) **are not compatible** with our work because they are designed for global concept discovery, learning fixed dictionary features across millions of activations [4]. These concepts do not adapt per image. In contrast,  to produce explanations, ClusCAM requires per-input grouping, which is performed at inference time without training or learning dataset-level concepts. However, using SAEs for concept-level explanations is an interesting direction for future work, though it would represent a different explanation paradigm rather than a replacement for CAM-style attribution.
> > ___
> > **[Q3. Normalization and upsampling]**
> >
> > Normalization is performed by **linearly scaling** each feature map to [0,1], i.e.,
> >
> > $\mathrm{Norm}(A) = \frac{A - \min(A)}{\max(A) - \min(A)}$, where A is a activation map.
> >
> > For upsampling, we use **bilinear interpolation**.
> >
> > These preprocessing choices follow the standard CAM pipeline: upsampling allows activation maps, originally defined on low-resolution feature grids, to be aligned with the input image and used meaningfully as perturbation masks. Likewise, normalizing each activation map to [0,1] produces smooth masks and addresses the inherent scale differences across activation maps. This avoids the instability and loss of information associated with binary masking. We will clarify these details in the revised manuscript.

---

> ### Author Response · Authors · 2025-11-21
>
> **[Q4. Why does M_j encode importance?]**
>
> The magnitude of M_j reflects how strongly the underlying internal representations in that group are activated at each spatial location. This aligns with the standard CAM intuition that higher activations correspond to regions contributing more strongly to the model’s internal computation.
>
> Importantly, we **do not** treat M_j itself as the importance measure. The **actual importance** comes from the logit-difference score, while M_j serves only as a spatial selector to test the functional effect of the group it represents. We will make these points explicit in the revised manuscript.
> ___
> **[Q5. Why dropout for filtering? Potential randomness / outlier issue]**
>
> **Why dropout for filtering.** As stated in Sec. 3.3, dropout in ClusCAM is motivated by the fact that some meta-representations decrease the target-class logit and therefore act as noisy groups. Filtering out these irrelevant groups helps improve the final explanation. This is supported by: (i) Figure 7 in the Supplementary shows that several groups have large negative scores and focus on noisy regions; (ii) the ablation study (Sec. 4.4, lines 454-463) demonstrates that removing dropout leads to a 23.37% drop in IC and an 18.54% increase in AD, confirming that retaining the noisy groups directly harms attribution quality.
>
> Additionally, we use the term “dropout” only because it matches the intuition of removing low-signal groups. We agree that this term may be confusing with conventional dropout that we often see in deep learning training pipelines, and we will consider changing it in the revised version.
>
> **Randomness of “Dropout”.** “Dropout” here is deterministic filtering, **not random sampling**: we remove the lowest r% scoring meta-representations (those with negative or near-zero logit differences). No randomness is used, and inference is fully deterministic.
>
> **Outlier heads in ViTs.** Recent work indeed links certain outlier behaviors to register tokens, and we acknowledge this line of research. Outlier detection in attention heads is an interesting potential extension, but it is **out of the scope of this work** (ClusCAM operates on activation maps or patch embeddings rather than attention heads).
> ___
> **[General summary]**
>
> *“In general, … using your approach.”*
>
> In summary, we **have provided brief and principled intuitions** for key components of ClusCAM (as a response to W2), addressing both the conceptual foundation and the empirical design choices (as a response to W3). Our method is built upon a **novel** formulation of group-wise attribution.
>
> We have justified each step with both theoretical reasoning and controlled ablations, clarifying how clustering, deterministic filtering, and temperature scaling work together, and we commit to making clarifications in the revised manuscript to strengthen the readability and accessibility of the paper.
>
> If you have any further concerns, we are pleased to respond.
> ___
> **References:**
>
> [1] Leem, S., & Seo, H. (2024, March). Attention guided CAM: visual explanations of vision transformer guided by self-attention. In Proceedings of the AAAI conference on artificial intelligence (Vol. 38, No. 4, pp. 2956-2964).
>
> [2] Mathilde Caron, Piotr Bojanowski, Armand Joulin, and Matthijs Douze. Deep clustering for unsupervised learning of visual features. In Proceedings of the European conference on computer vision (ECCV), pp. 132–149, 2018.
>
> [3] Mathilde Caron, Ishan Misra, Julien Mairal, Priya Goyal, Piotr Bojanowski, and Armand Joulin. Unsupervised learning of visual features by contrasting cluster assignments. Advances in neural information processing systems, 33:9912–9924, 2020.
>
> [4] Wang, H., Wang, Z., Du, M., Yang, F., Zhang, Z., Ding, S., ... & Hu, X. (2020). Score-CAM: Score-weighted visual explanations for convolutional neural networks. In Proceedings of the IEEE/CVF conference on computer vision and pattern recognition workshops (pp. 24-25).
>
> [5] Byun, S. Y., & Lee, W. (2024). ReciproCAM: Lightweight Gradient-free Class Activation Map for Post-hoc Explanations. In Proceedings of the IEEE/CVF Conference on Computer Vision and Pattern Recognition (pp. 8364-8370).
>
> [6] Ramaswamy, H. G. (2020). Ablation-cam: Visual explanations for deep convolutional network via gradient-free localization. In proceedings of the IEEE/CVF winter conference on applications of computer vision (pp. 983-991).
>
> [7] Cunningham, Hoagy, et al. "Sparse autoencoders find highly interpretable features in language models." arXiv preprint arXiv:2309.08600 (2023).

---

> > ### Comment · Reviewer_rjyS · 2025-11-22
> >
> > Thanks for your thorough explanations and additional experiments. Here is my feedback:
> >
> > [W2. Intuitive explanation for each step]
> >
> > Regarding Kmeans++ (and other sections), your intuition as I understand it is explaining why general clustering is needed, not specifically K-means. The ablation on other clustering methods is more convincing, although it put more focus on the intuition behind heuristics, which my intention was on the general conceptual contribution. (you meant 426-455 according to the given submission).
> >
> > [W.3 Conceptual vs. empirical contributions]
> >
> > This is my main concern with your approach, the idea of splitting into "groups" or concepts in order to enhance explainability is a reasonable approach, and is widely explored. The small details in my opinion, are hiding the conceptual contribution rather than supporting it although they enhance the performance. If the main idea is as you mentioned, then you can use existing approaches to split into groups i.e., [1] (there are many more).  Put the main contribution at the center core of your approach, and show that this is the main contributor for explanations, and all the other tweaks are enhancing a bit. Currently, in the way the paper written, it put too much weight on heuristics.
> >
> > [Q2. Clustering vs. Sparse Autoencoders]
> >
> > There are plenty of methods to decompose single given image into concepts, please see for example [2], this is a very growing literature.
> >
> > [Q5. Why dropout for filtering? Potential randomness / outlier issue]
> >
> > I think that the term dropout is misleading, and it is better to rename it. Moreover, the whole system of filtering seems also heuristic to me (how much to filter, maybe not filter only the r% least important but other combination and more), and you should try and discuss the "registers" paper [3, in my main review], because it is very related to what you are trying to filter here.
> >
> >
> > Again, I appreciate the authors' effort and additional experiments and explanations, however I still think that the method should focus on the main contribution in terms of writing and experimenting and less on validating and justifying empirical decisions. If possible, to lean on given approaches to isolate the essence of the method. This will make your method be more clean and clear, and pave the way more easily to others to extend it.
> >
> > refs:
> >
> > Interpreting CLIP with Sparse Linear Concept Embeddings (SpLiCE). Usha Bhalla et al. Neurips 2024.
> >
> > Quantifying Structure in CLIP Embeddings: A Statistical Framework for Concept Interpretation. Zhao et al. arxiv.

---

> > > ### Author Response · Authors · 2025-12-01
> > >
> > > **[W2. Intuitive explanation for each step]**
> > >
> > > Thank you again for this point. We agree that the paper should be rewritten, as clustering is a conceptual contribution, and K-means++ is just a suitable tool to do this. We updated this in the revised version.
> > >
> > > **[W.3 Conceptual vs. empirical contributions]**
> > >
> > > We agree that the previous manuscript strongly focuses on implementation details, which makes the core conceptual contribution unclear. In the revised version, we have restructured the methodology section to center on the proposed clustering mechanism as the primary driver for explainability.
> > >
> > > **[Q2. Clustering vs. Sparse Autoencoders]**
> > >
> > > We thank the reviewer for highlighting this growing line of work.
> > >
> > > In the XAI domain, this research direction falls under **concept-based explanations**, where the goal is to identify interpretable concepts (e.g, texture patterns, color gradients, object parts,...)  and quantify their individual contributions to the model’s predictions. To the best of our knowledge, the first work that explicitly formulates this problem as both concept identification and concept attribution was introduced in 2025 [1], which formalizes the process of extracting concepts and measuring their influence on model outputs.
> > >
> > > In contrast, our work lies in **attribution-based explanations**, which focus on directly measuring how much each internal representation (i.e, feature maps or patch tokens), instead of concepts, contributes to the final prediction. More specifically, the proposed method, ClusCAM,  aims at improving the faithfulness of CAM-style attribution-based explanations by mitigating the influence of spurious or low-informative clusters.
> > >
> > > Therefore,  although concept-based explanations are growing quickly, they should be deemed complementary rather than competitive with their attribution-based counterparts in general, and ClusCAM in particular.
> > > In the future, integrating concept-based explanation methods, such as sparse autoencoders with CAM attribution-based explanation methods, can be promisingly investigated.
> > >
> > > [1] Kuroki, M., & Yamasaki, T. (2025). CE-FAM: Concept-Based Explanation via Fusion of Activation Maps. In Proceedings of the IEEE/CVF International Conference on Computer Vision (pp. 1413-1422).
> > >
> > > **[Q5. Why dropout for filtering? Potential randomness / outlier issue]**
> > >
> > > We thank the reviewer for this insightful comment.
> > >
> > > First, we agree that the term “dropout” may feel confusing, and we will then rename it to a more comprehensible term: discarding.
> > >
> > > Second, regarding our discarding strategy, we aim to avoid introducing additional trainable components or heavy optimization at inference time. We will revise the manuscript to make this motivation clearer.
> > >
> > > Finally, concerning the relation to the register tokens paper [3], both works start from **a similar observation**: certain patch tokens in ViTs tend to carry little local information while being repurposed for global computation, which negatively affects interpretability. However, the two approaches tackle this issue at different levels. The register method proposed in [3] addresses the problem at the architecture and representation learning stage by **introducing auxiliary tokens during training**, whereas our method operates at the explanation stage, **without modifying the backbone or requiring retraining**. We thus view these approaches as complementary. In fact, combining register-based architectures with CAM-style explanation methods is a very promising direction for improving the explainability of ViTs.
> > >
> > > Once again, thank you so much for your response. We hope that these will address your concerns.

---

### Official Review · Reviewer_ztRk · 2025-10-26

**Soundness:** 2
**Presentation:** 3
**Contribution:** 3
**Rating:** 4
**Confidence:** 4

**Summary:**

This paper proposes ClusCAM, a novel gradient-free post-hoc explanation framework designed to enhance the faithfulness and interpretability of visual explanations in image classification models. Unlike conventional CAM-based approaches that treat internal representations as independent, ClusCAM clusters them into semantically coherent meta-representations using the K-Means++ algorithm. The importance of each cluster is quantified through logit-based differences, followed by a dropout mechanism and temperature-scaled softmax to suppress irrelevant signals and highlight the most influential regions. ClusCAM is architecture-agnostic, effectively applicable to both convolutional neural networks (CNNs) and Vision Transformers (ViTs). Extensive experiments show that ClusCAM consistently outperforms state-of-the-art baselines across multiple quantitative metrics, producing sharper and more interpretable visualizations.

**Strengths:**

ClusCAM introduces a group-wise attribution strategy by clustering internal representations into higher-level meta-representations. This approach marks a significant improvement over conventional CAM methods, which assume that individual features contribute independently and with equal importance—often resulting in noisy or unreliable explanations. Furthermore, the paper presents a data-driven procedure for selecting key hyperparameters, thereby reducing the reliance on manual tuning and improving the overall stability and reproducibility of the method.

**Weaknesses:**

ClusCAM introduces additional computational overhead compared to highly efficient methods such as Grad-CAM. The initial K-Means++ clustering of internal representations increases inference time, particularly for large-scale models. Although ClusCAM can operate faster than exhaustive ablation-based approaches like Score-CAM and Ablation-CAM when applied to Vision Transformers (ViTs), its computational cost still limits scalability in real-time or resource-constrained environments. Moreover, while the paper proposes data-driven strategies for selecting key hyperparameters, these procedures are primarily heuristic and lack a strong theoretical foundation, leaving room for further formal analysis and optimization.

**Questions:**

While the proposed method demonstrates several notable strengths, I have some concerns regarding its broader applicability and theoretical grounding. For instance, gradient-based analyses still provide valuable information for enhancing output probabilities, as they capture model sensitivity through backward propagation. It would be worthwhile to investigate whether ClusCAM could be integrated with gradient-based interpretability approaches, since many state-of-the-art explanation frameworks leverage both forward and backward reasoning. Although the reported results indicate that ClusCAM outperforms several contemporary baselines, a more comprehensive comparison with recent state-of-the-art model, such as Attention-Guided CAM (AAAI 2024) combining forward and backward attention mechanisms to suppress noise in Vision Transformer, would further strengthen the empirical validation of this work. Additionally, for small target objects, the hyperparameter, particularly the number of clusters, may have a significant impact on the interpretability and stability of the resulting explanations, and this sensitivity warrants further analysis. For the temperature-scaled softmax, ClusCAM uses a τ value less than one. I agree that without temperature scaling, the softmax weights can become overly uniform. However, a low τ amplifies noisy or erroneous signals. Therefore, it would be beneficial to validate this behavior using test images that contain large homogeneous backgrounds with small target objects.

---

> ### Author Response · Authors · 2025-11-21
>
> Dear Reviewer ztRk,
>
> Thank you for the constructive feedback. Regarding the weaknesses, we would like to further explain in detail:
> ___
> **[W1. Computational overhead]**
>
> *“ClusCAM introduces … resource-constrained environments.”*
>
> The goal of ClusCAM does not focus on *“real-time or resource-constrained environments”*, but rather to serve as a faithful gradient-free, attribution-based explanation method. Hence, we want to clarify two points:
>
> **(1) We trade runtime for attribution accuracy**, which is a common pattern across most gradient-free explanation methods (e.g., [1, 2]). Besides Figure 6 (Average running times) in the manuscript, we report the trade-off in the following table.
>
> *Table 1. Trade-off analysis: multiplicative gains in performance and runtime of ClusCAM and the two best gradient-free methods compared to GradCAM.*
> | Arch | Method   | ScoreCAM | OptiCAM | ClusCAM |
> |------|----------|----------|---------|---------|
> | CNNs | AD       | x1.29    | x2.16   | **x2.42** |
> | CNNs | IC       | x1.37    | x1.19   | **x1.7**  |
> | CNNs | AG       | x1.52    | x0.81   | **x1.92** |
> | CNNs | Runtime  | **x23.43** | x34.11 | x44.24 |
> | ViTs | AD       | x1.36    | **x18.61** | x13.6 |
> | ViTs | IC       | x3.27    | x9.13   | **x12.1** |
> | ViTs | AG       | x4.3     | x5.59   | **x19.39** |
> | ViTs | Runtime  | x24.9    | x43.69  | **x22.16** |
>
> It is clear that while ClusCAM has a higher runtime on CNNs (x44.24), this increase from the intentional clustering step (see Figure 6) is proportional to the gains in performance (x2.42 AD, x1.7 IC, and x1.92 AG) and is not excessively large relative to the two typical gradient-free CAM baselines. Notably, this effect diminishes on ViTs, where ClusCAM achieves the lowest runtime among the three methods (x22.16) but exhibits the best IC and AG scores (i.e, x12.1 and x19.39, respectively).
>
> **(2) To mitigate the cost of this trade-off:** (a) We **keep the efficiency of the scoring phase of ClusCAM** to make it competitive among the gradient-free CAM methods. For K << N, the number of forward passes (FP) required in some typical explanation baselines is shown in the table below:
>
> *Table 2. The number of forward passes (FP) required in some typical explanation baselines (K<<N).*
> | Method        | GradCAM | GradCAM++ | ScoreCAM | AblationCAM | ClusCAM |
> |---------------|---------|-----------|----------|-------------|---------|
> | Number of FP  | 1       | 1         | N        | N           | K       |
>
> It is clear that the number of FP of ClusCAM is significantly smaller than ScoreCAM and AblationCAM (e.g., on ResNet18, K = 30 and N = 512).
>
> (b) Moreover, the **K-means++ phase on ViTs is more efficient than that on CNNs** because the number of internal representations decreases (e.g., 512 with ResNet18 but only 196 with ViT-B).
> We have mentioned it in lines 956 to 962. We will make this trade-off clearer by adding the tables and discussions in the revised manuscript.
>
> We hope these points help clarify your concern.
> ___
> **[W2. Hyperparameter selection]**
>
> *“Moreover, while the paper … formal analysis and optimization.”*
>
> While our selection procedures are data-driven, they **are not heuristic** in the arbitrary sense. More specifically:
>
> * Elbow-based selection [3] for K stems from classical clustering theory, where curvature changes reflect diminishing returns in intra-cluster variance reduction. This criterion has been widely adopted as a theoretically motivated proxy [4].
>
> * The GMM-based estimation [5] of r relies on mixture-model posterior probabilities, a standard statistical tool for separating signal and noise components [6].
>
> * The temperature τ is defined analytically as a function of (r, K), ensuring well-behaved scaling of softmax sharpness, avoiding arbitrary choices.
>
> We mentioned these theoretical motivations in Appendix. B. Selecting hyperparameters. We will highlight these in the main text to make the revised version as clear as possible.

---

> > ### Author Response · Authors · 2025-11-21
> >
> > We also address the reviewer’s questions below:
> > ___
> > **[Q1. Integrating gradients]**
> >
> > *“For instance, gradient-based analyses … forward and backward reasoning.”*
> >
> > Thank you for this suggestion. However, we would like to clarify that this combination is feasible, but **out of the scope of this work**. This is because:
> >
> > **Our current gradient-free design is intentional:** for CNNs, gradient-based CAM methods suffer from well-documented limitations, including saturation, false confidence, and constraints in post-deployment settings (lines 114 to 119; Sec. 2). This trend is also reflected in many recent CAM approaches, and even OptiCAM (which is designed for both CNNs and ViTs) has shifted toward gradient-free.
> >
> > However, **ClusCAM is compatible with gradient-based attribution** and can readily be extended by replacing the logit-difference score (see Eq. 5) with gradient-derived signals aggregated at the cluster level (e.g., ∂y/∂F averaged within each meta-representation or attention-flow–based gradients for ViTs). This enables a hybrid “forward + backward” group-wise attribution, which represents an extension to future work.
> > ___
> > **[Q2. Comparing with Attention-Guided CAM for ViTs]**
> >
> > *“Although the reported … empirical validation of this work.”*
> >
> > We agree with the reviewer that **including an Attention-Guided CAM [7] method would further strengthen our work**. We experimented and reported the results below:
> >
> > *Table 3. Quantitative comparison of explanation methods on ViT-B.*
> > | Method       | AD ↓  | IC ↑  | AG ↑  |
> > |-------------|-------|-------|-------|
> > | GradCAM     | 73.40 | 7.20  | 4.54  |
> > | GradCAM++   | 74.46 | 7.45  | 5.32  |
> > | ScoreCAM    | 56.03 | 19.10 | 15.50 |
> > | AblationCAM | 57.94 | 15.25 | *20.14* |
> > | ReciprocCAM | 53.46 | 9.00  | 8.68  |
> > | OptiCAM     | **4.69**  | *36.30* | 13.12 |
> > | ShapleyCAM  | 74.44 | 5.21  | 1.22  |
> > | ClusCAM     | *5.21* | **60.75** | **73.96** |
> > | AG-CAM      | 17.61 | 22.35 | 6.28  |
> >
> > Note that AG-CAM is specifically designed for the traditional ViT architecture, while our method **ClusCAM works well with both ViT and its variants (e.g, Swin, LeViT, CaiT, PVT)**. The results show that AG-CAM outperforms most methods that were originally developed for CNN-based models. However,  compared with approaches explicitly optimized for transformer-based models, such as OptiCAM and ClusCAM, AG-CAM performs much less favorably. We will clarify this distinction in the revision.

---

> > > ### Author Response · Authors · 2025-11-21
> > >
> > > **[Q3. Sensitivity analysis for small target objects]**
> > >
> > > *“Additionally, for small target objects, … sensitivity warrants further analysis.”*
> > >
> > > ClusCAM’s stability **may not depend on object size** but on the general clustering behavior of internal representations.
> > >
> > > Prior work shows that internal features organize according to semantic and visual similarity in the activation space [8, 9], and this property naturally extends to small-object cases. In such settings, object-related feature maps typically form only a small subset of all channels (e.g., 100 out of 512), while most background-dominated maps (412) cluster separately. Since K-means++ groups by similarity rather than quantity, object-related maps remain coherent instead of being diluted.
> > >
> > > Moreover, background-heavy clusters yield near-zero or negative logit differences and are suppressed by the dropout step of ClusCAM, effectively preventing them from influencing the final explanation. Empirically, ClusCAM shows stable IC/AG performance on both CNNs and ViTs over the full ImageNet benchmark (Table 5), which naturally includes images containing small target objects such as birds, insects, and bottle caps.
> > > ___
> > > **[Q4. Temperature-scaled softmax for small target objects]**
> > >
> > > *“For the temperature-scaled softmax … small target objects.”*
> > >
> > > Temperature scaling is **not expected to introduce instability for small target objects** because it is applied right **after** the dropout step.
> > >
> > > By this point, irrelevant groups have already been discarded, and τ simply sharpens the distribution over the remaining high-confidence clusters. Since this mechanism is derived from the group-wise scoring and holds for all images regardless of object size, it inherently applies to small-object cases as well. Thus, τ < 1 **does not amplify noise** but reinforces truly salient meta-representations.
> > >
> > > **Summary for Q3 and Q4 (small target objects).** Although our analysis and empirical results suggest that ClusCAM’s stability and temperature scaling behavior are mostly independent of object size (due to its reliance on clustering over internal representations rather than spatial extent),  we agree with the reviewer that an analysis of the sensitivity of the hyperparameters for a special case of small target objects could further strengthen this work. Unfortunately, identifying a suitable large-scale benchmark with reliably annotated small target objects is non-trivial.
> > >
> > > ___
> > >
> > > **References:**
> > >
> > > [1] Wang, H., Wang, Z., Du, M., Yang, F., Zhang, Z., Ding, S., ... & Hu, X. (2020). Score-CAM: Score-weighted visual explanations for convolutional neural networks. In Proceedings of the IEEE/CVF conference on computer vision and pattern recognition workshops (pp. 24-25).
> > >
> > > [2] Zhang, H., Torres, F., Sicre, R., Avrithis, Y., & Ayache, S. (2024). Opti-CAM: Optimizing saliency maps for interpretability. Computer Vision and Image Understanding, 248, 104101.
> > >
> > > [3] Thorndike, R. L. (1953). Who belongs in the family?. Psychometrika, 18(4), 267-276.
> > >
> > > [4] Shi, C., Wei, B., Wei, S., Wang, W., Liu, H., & Liu, J. (2021). A quantitative discriminant method of elbow point for the optimal number of clusters in clustering algorithm. EURASIP journal on wireless communications and networking, 2021(1), 31.
> > >
> > > [5] Melchior, P., & Goulding, A. D. (2018). Filling the gaps: Gaussian mixture models from noisy, truncated or incomplete samples. Astronomy and computing, 25, 183-194.
> > >
> > > [6] Melchior, P., & Goulding, A. D. (2018). Filling the gaps: Gaussian mixture models from noisy, truncated or incomplete samples. Astronomy and computing, 25, 183-194.
> > >
> > > [7] Leem, S., & Seo, H. (2024, March). Attention guided CAM: visual explanations of vision transformer guided by self-attention. In Proceedings of the AAAI conference on artificial intelligence (Vol. 38, No. 4, pp. 2956-2964).
> > >
> > > [8] Caron, M., Bojanowski, P., Joulin, A., & Douze, M. (2018). Deep clustering for unsupervised learning of visual features. In Proceedings of the European conference on computer vision (ECCV) (pp. 132-149).
> > >
> > > [9] Caron, M., Misra, I., Mairal, J., Goyal, P., Bojanowski, P., & Joulin, A. (2020). Unsupervised learning of visual features by contrasting cluster assignments. Advances in neural information processing systems, 33, 9912-9924.

---

> ### Comment · Reviewer_ztRk · 2025-11-25
>
> Thank you for taking the time to address my concerns. I appreciate the clarification that ClusCAM can theoretically incorporate both forward and backward attribution, and the additional experiments you provided do help substantiate parts of the contribution. The method’s theoretical foundations are well-motivated and coherent, and the new results further strengthen certain claims. However, important concerns remain regarding practicality and empirical completeness. Even if real-time processing is not the goal, the computational overhead—especially the reported 44× slowdown on CNNs—still appears excessively large. The rebuttal frames this as an accuracy–runtime trade-off, yet provides no absolute latency measurements, cost–benefit justification, or exploration of lighter clustering alternatives, making it difficult to judge feasibility in broader interpretability workflows.
>
> The responses on small objects and multiple-instance settings are also largely conceptual. While clustering in activation space may offer some robustness, the absence of experiments that explicitly isolate small or weak objects leaves the claims unverified. The natural diversity of ImageNet does not replace controlled evaluations across object sizes or densities, and it remains unclear how consistently ClusCAM preserves small or spatially fragmented signals under dropout.
>
> Similarly, the explanation of the temperature-scaled softmax lacks empirical evidence. Although τ is described as merely sharpening the remaining scores, potential edge cases—such as sparse or low-magnitude clusters—are not examined, and τ < 1 could amplify noise. Sensitivity analyses under such challenging scenarios would help validate stability.
>
> Finally, design aspects such as per-image clustering, hyperparameter selection, and a reliance on confidence-based metrics raise additional questions about scalability and robustness. Overall, while ClusCAM is conceptually appealing and the intentions are clear, further empirical validation—particularly regarding computational cost and edge-case behavior—would substantially strengthen the work.
>
> Once again, I appreciate the effort the authors have put into responding to my comments, and I believe that adequately addressing the points raised here will meaningfully improve the quality of the paper.

---

> > ### Author Response · Authors · 2025-11-28
> >
> > Thank you to the reviewer for your feedback. We would again highlight the trade-off compared to OptiCAM (the second-best baseline) to show the benefits of ClusCAM. The results show that ClusCAM significantly outperforms OptiCAM in most aspects. On ViTs, improving up to 3.47 times in AG with only 0.51 times in runtime is the strongest aspect of ClusCAM. On CNNs, ClusCAM only increases 1.31 times in runtime, but improves up to 2.37 times in AG.
> >
> > | Arch | Method   | ClusCAM / OptiCAM |
> > |------|----------|--------------------|
> > | CNNs | AD       | **1.12×**              |
> > | CNNs | IC       | **1.43×**              |
> > | CNNs | AG       | **2.37×**              |
> > | CNNs | Runtime  | 1.30×              |
> > | ViTs | AD       | **0.73×**              |
> > | ViTs | IC       | **1.33×**              |
> > | ViTs | AG       | **3.47×**             |
> > | ViTs | Runtime  | **0.51×**             |
> >
> > Moreover, we agree with the reviewer that **the special experiment of small objects** and **the deep analysis of hyperparameters** are **interesting future work**, which would substantially strengthen our work. These will be included in the extended version of ClusCAM.
> >
> > Once again, thank you so much for your comments.

---

### Official Review · Reviewer_PgLt · 2025-10-30

**Soundness:** 3
**Presentation:** 3
**Contribution:** 3
**Rating:** 6
**Confidence:** 4

**Summary:**

This paper propose ClusCAM, a gradient-free post-hoc explanation method that groups internal representations into meaningful clusters (meta-representations). The importance of each cluster is then measured through logit differences with dropout and temperature-scaled softmax, emphasizing the most influential groups.

By modeling group-wise interactions, ClusCAM generates sharper, more interpretable, and faithful explanations. The method is architecture-agnostic, working with both CNNs and Vision Transformers. Experimental results show that ClusCAM surpasses state-of-the-art interpretability techniques.

**Strengths:**

The paper proposes a new interpretability method, ClusCAM, and provides extensive experimental validation to demonstrate its effectiveness. The authors present numerous quantitative results that highlight the superiority of their approach.

The experimental setup is highly comprehensive, covering multiple datasets—including the ILSVRC2012 benchmark and an Alzheimer’s MRI dataset—and a wide range of model backbones, such as ResNet variants (ResNet-18/34/50/101), EfficientNet, InceptionNet, and various Vision Transformers (e.g., ViT-B, Swin-B, LeViT-192/256, CaiT-XXS-24, and PVTv2).

A diverse set of evaluation metrics is also employed to thoroughly demonstrate the robustness and effectiveness of the proposed method.

**Weaknesses:**

The paper presents extensive experiments across multiple backbones to demonstrate the superiority of ClusCAM, but it lacks comparisons with several important baseline methods:
1. Vitali Petsiuk, Abir Das, and Kate Saenko. RISE: randomized input sampling for explanation of black-box models. In British Machine Vision Conference 2018, BMVC 2018, Northumbria University, Newcastle, UK, September 3-6, 2018, page 151, 2018.
2. Quan Zheng, Ziwei Wang, Jie Zhou, and Jiwen Lu. 2022. Shap-CAM: Visual Explanations for Convolutional Neural Networks Based on Shapley Value. In Computer Vision–ECCV 2022: 17th European Conference. Springer, Tel Aviv, Israel, 459–474

In addition to proposing ClusCAM, which is extensively validated across multiple backbones to demonstrate its superiority, the paper offers limited insight into the underlying model behavior. Several prior studies have explored interpretability and explanation in deep models from different perspectives.
1. Rulin Shao, Zhouxing Shi, Jinfeng Yi, Pin-Yu Chen, and Cho-Jui Hsieh. On the adversarial robustness of visual transformers. arXiv preprint arXiv:2103.15670, 2021.
2. Yutong Bai, Jieru Mei, Alan L Yuille, and Cihang Xie. Are transformers more robust than cnns? Advances in Neural Information Processing Systems, 34, 2021.
3. Mingqi Jiang, Saeed Khorram, and Li Fuxin. Comparing the decision-making mechanisms by transformers and cnns via explanation methods. In IEEE Conf. Comput. Vis. PatternRecog. (CVPR), pages 9546–9555, 2024.

**Questions:**

Are there any future plans to extend or apply this method to other tasks or domains?

---

> ### Author Response · Authors · 2025-11-21
>
> Dear Reviewer PgLt,
>
> We thank the reviewer for the valuable and constructive feedback. Below, we target each of your concerns one by one.
> ___
> **[W1. Adding baselines]**
>
> Our evaluation follows the standard comparison used in recent CAM-style literature (e.g, [1, 2, 3]). We do not include RISE due to its consistently reported weak performance, and exclude Shap-CAM because its source code is unfortunately not available. More specifically:
>
> **Comparison vs. RISE (Petsiuk et al., 2018).** RISE seems **no longer treated as a main baseline** in the most recent works [1,2,3]. It is also reported in [4] to perform considerably worse than common CAM-based methods on CNNs. However, following the reviewer’s suggestion, we have performed experiments to validate the performance of RISE compared to other methods. The results in Tables 1 and 2 show that RISE is not competitive with the others on CNNs. On ViTs, RISE’s performance is more comparable, but still underperforms ClusCAM. We will add these results to the revised version.
>
> *Table 1. Evaluation of ClusCAM and RISE across 6 different CNN-based architectures.*
> | Model         | ClusCAM AD ↓ | ClusCAM IC ↑ | ClusCAM AG ↑ | RISE AD ↓ | RISE IC ↑ | RISE AG ↑ |
> |---------------|--------------|--------------|--------------|-----------|-----------|-----------|
> | ResNet-18     | **11.50** | **50.50** | **22.99** | 22.05 | 34.75 | 15.24 |
> | ResNet-34     | **9.16**  | **57.50** | **32.06** | 19.40 | 38.35 | 13.94 |
> | ResNet-50     | **8.58**  | **57.20** | **32.19** | 16.83 | 39.35 | 12.11 |
> | ResNet-101    | **6.57**  | **61.50** | **36.37** | 57.26 | 9.40  | 2.06  |
> | EfficientNet  | **4.84**  | **67.15** | **28.36** | 46.18 | 17.85 | 6.55  |
> | InceptionV3   | **6.29**  | **63.60** | **46.64** | 58.53 | 10.60 | 2.05  |
>
> *Table 2. Evaluation of ClusCAM and RISE across 6 different ViT-based architectures.*
> | Model         | ClusCAM AD ↓ | ClusCAM IC ↑ | ClusCAM AG ↑ | RISE AD ↓ | RISE IC ↑ | RISE AG ↑ |
> |---------------|--------------|--------------|--------------|-----------|-----------|-----------|
> | ViT-B         | **5.21**  | **60.75** | **73.96** | 20.79 | 22.10 | 6.77  |
> | Swin-B        | **8.22**  | **22.90** | **45.49** | 59.43 | 5.65  | 1.02  |
> | LeViT-192     | **1.33**  | **80.25** | **20.30** | 43.32 | 12.05 | 2.92  |
> | LeViT-256     | **1.55**  | **74.75** | **15.52** | 33.71 | 12.70 | 2.38  |
> | CaiT-XXS-24   | **5.51**  | **41.80** | **15.38** | 42.58 | 8.00  | 2.51  |
> | PVTv2         | **12.03** | **47.80** | **16.66** | 61.92 | 7.75  | 3.53  |
>
> **Comparison vs. Shap-CAM (Zheng et al., 2022).** We do not include Shap-CAM due to the **lack of a publicly available implementation**, which prevents reliable reproduction. Moreover, we have already included Shapley-CAM (2025) [3] in our experiments (Tab. 2 in Sec. 4.2. Quantitative analysis). As this method is also grounded in Shapley value theory, we believe it serves as an up-to-date representative. Therefore, we hope that this inclusion adequately addresses the reviewer’s concern regarding Shapley-based comparisons. We will clarify these points in the revised manuscript.
> ___
> **References**
>
> [1] Byun, S. Y., & Lee, W. (2024). ReciproCAM: Lightweight Gradient-free Class Activation Map for Post-hoc Explanations. In Proceedings of the IEEE/CVF Conference on Computer Vision and Pattern Recognition (pp. 8364-8370).
>
> [2] Zhang, H., Torres, F., Sicre, R., Avrithis, Y., & Ayache, S. (2024). Opti-CAM: Optimizing saliency maps for interpretability. Computer Vision and Image Understanding, 248, 104101.
>
> [3] Cai, H., Yang, Y., Tang, Y., Sun, Z., & Zhang, W. (2025). Shapley value-based class activation mapping for improved explainability in neural networks. The Visual Computer, 1-19.
>
> [4] Wang, H., Wang, Z., Du, M., Yang, F., Zhang, Z., Ding, S., ... & Hu, X. (2020). Score-CAM: Score-weighted visual explanations for convolutional neural networks. In Proceedings of the IEEE/CVF conference on computer vision and pattern recognition workshops (pp. 24-25).

---

> > ### Author Response · Authors · 2025-11-21
> >
> > **[W2. Insight into underlying model behavior of CNNs and ViTs]**
> >
> > We appreciate the reviewer’s interest in analyzing the underlying behaviors of CNNs and ViTs. While this is indeed an important direction, it falls **outside the scope of this paper**.
> >
> > ClusCAM is an attribution-based explanation method that aims to identify which **input regions most strongly influence the model’s prediction**. Meanwhile, *(Shao et al., 2021)* and *(Bai et al., 2021)* studied **the robustness** of ViTs against adversarial perturbations compared to CNNs. *(Jiang et al., 2024)* proposed two new methodologies: subexplanation counting and cross-testing to differentiate the **decision-making behaviors of ViTs and CNNs**. These works focus on **different aspects of model behavior** and thus are beyond the scope of our attribution-based study.
> >
> > We therefore leave a deeper investigation of the behavior of CNN vs. ViT models for future work. We will highlight this point in the discussion part of our paper.
> > ___
> > **[Q1. Future plans]**
> >
> > Yes, we see that there are several potential extensions. Specifically:
> >
> > **(i) Extending to different tasks**, such as object detection and semantic segmentation, where grouping spatial features may help disentangle object-context interactions, as mentioned in lines 968 to 971 in the manuscript;
> >
> > **(ii) 3D-image models** (e.g., volumetric medical imaging or video-based architectures), where clustering could incorporate spatio-temporal consistency;
> >
> > **(iii) Insight into underlying model behavior**, as suggested in W2. ClusCAM may facilitate analyses of robustness or architectural differences between CNNs and ViTs by revealing which features the model consistently relies on. If a model systematically focuses on global structures vs. local textures, this pattern reflects its inductive bias.
> > ___
> > Finally, we would be happy to address any additional questions you may have.

---

### Author Response · Authors · 2025-12-04

Dear Reviewers and ACs,

___

We thank all reviewers for their constructive feedback. Throughout the rebuttal, we have addressed the major concerns raised regarding *empirical completeness, conceptual clarity, and methodological explainability*. We now summarize the key clarifications below:

**1. Strengthened experimental evaluation**

*[As suggested by Reviewer PgLt]*

We expanded our benchmark to include additional baselines: added **RISE** and **Attention-Guided CAM** results, showing that ClusCAM still outperforms these methods, further strengthening our work. More specifically:
- RISE underperforms ClusCAM at all metrics;
- AG-CAM is better than CNN-based explanation methods on ViT, but still underperforms OptiCAM and ClusCAM.

**2. Clarified computational efficiency**

*[As suggested by Reviewer ztRk]*

- Explained the accuracy–runtime trade-off inherent to **gradient-free** attribution methods.
- Highlighted strong trade-offs compared to the second-best baseline (OptiCAM):
  - **ViTs**: up to **3.47×** AG improvement with **0.51× runtime**
  - **CNNs**: up to **2.37×** AG improvement with **1.30× runtime**
- Reduced forward passes to **K ≪ N**, significantly lighter than ScoreCAM/AblationCAM.
- Clarified ViT efficiency benefits due to fewer internal representations.
- Included runtime tables & discussion in the revision.

**3. Enhanced conceptual contribution**

*[As suggested by Reviewer rjyS]*

To address concerns about “too many heuristics” diluting novelty:
- Rewrote the methodology section to **center the conceptual contribution**:
  ➜ **Group-wise attribution** as a new explanatory principle.
- K-means++, filtering, and temperature scaling are now positioned as **instantiations** of this core idea, not the idea itself.

We also emphasize that ClusCAM improves **faithfulness of CAM-style attribution**, and serves as a *complement*, not a competitor, to concept-based XAI.

**4. Improved intuitiveness of design choices**

*[As suggested by Reviewer rjyS]*

In order to make the pipeline more interpretable, we added clearer motivations for:
- K-means++ clustering (stable co-activation grouping)
- Logit-difference scoring (faithful functional contribution)
- **Discarding** filtering (renamed from “dropout”)
- Temperature modulation after noise suppression

**5. Clarified scope regarding CNN vs. ViT Behavior**

*[As suggested by Reviewer PgLt and ztRk]*

We state that analyzing underlying architectural behavior is **future work** beyond attribution scope.ClusCAM could serve as a tool for such analysis in follow-up studies.

**6. Planned directions**

*[As suggested by Reviewer PgLt and  rjyS]*

We agree and plan to explore:
- Using ClusCAM as a tool to differentiate CNN and ViT behaviors.
- Integration with hybrid forward–backward attribution to balance the benefits of gradient and gradient-free methods.
- Extending ClusCAM from attribution-based to concept-based explanation by combining concept-level models with group-wise attribution.

___

Once again, we appreciate the reviewers’ insights, which helped improve both the scientific clarity and presentation quality of our work. We believe the revised manuscript, with the *newly added experiments, methodological restructuring, clarified motivations, and roadmap for future extensions*, significantly enhanced the quality of our work.

---

### Meta-Review · Area_Chair_hAF3 · 2026-01-05

**Summary:**

Across reviewers, the common concerns center on limited conceptual novelty and insufficient theoretical grounding, incomplete empirical comparisons, and practical scalability issues.

(a) While ClusCAM demonstrates strong empirical performance across multiple backbones, reviewers consistently note that the core idea — grouping or clustering internal representations for improved explainability — largely overlaps with existing concept-based XAI methods, and that the paper places heavy emphasis on heuristic, empirically motivated design choices (e.g., K-means++, filtering/dropout, temperature scaling) without sufficiently isolating or clearly articulating the central conceptual contribution. Reviewers agree that the paper would benefit from stronger intuitive explanations and theoretical justification for each major step, and more targeted experiments to validate stability, robustness, and practicality beyond aggregate benchmarks.

(b) Several reviewers highlight missing or incomplete baselines, particularly recent state-of-the-art CAM, transformer-specific XAI methods, and alternative concept extraction approaches (e.g., SAE-based methods), making it difficult to fully contextualize the claimed advantages.

(c) Computational overhead and scalability: reviewers pointed out that per-image clustering introduces significant inference cost, with limited analysis of absolute latency, lightweight alternatives, or robustness in edge cases such as small objects or noisy activations.

**Reviewer Concerns:**

The authors provided additional benchmarking experiments against more recent CAM-based baselines, including methods specifically designed for Vision Transformers, which strengthens the empirical evidence that ClusCAM achieves improved AD, IC, and AG metrics. However, some evaluation details (e.g., datasets) remain insufficiently specified. The rebuttal also includes runtime and computational cost analyses, which largely address reviewers’ concerns regarding scalability.

Nevertheless, the core concerns regarding conceptual novelty remain only partially resolved. As noted by Reviewer rjyS, the proposed CAM-based pipeline relies on a sequence of heuristic design choices that lack strong theoretical or principled justification. While group-wise attribution is presented as the main contribution, its qualitative and quantitative advantages over existing concept-based explanation methods are not convincingly demonstrated in the rebuttal. In addition, although relatively secondary, the lack of targeted analysis on challenging scenarios — such as very small objects or highly cluttered backgrounds — limits the evidence for robustness and generalizability of the proposed explanations.

Overall, while the rebuttal improves the empirical positioning of the method, the remaining concerns regarding conceptual clarity, justification, and robustness suggest that the paper does not yet meet the acceptance threshold. I therefore recommend rejection at this stage.

**Reviewer Scores:**

I believe that all reviewers would likely maintain their original scores, as most actively participated in the post-rebuttal discussion and their primary concerns were either addressed or partially addressed in the authors’ rebuttal.

---

### Decision · Program_Chairs · 2026-01-26

Reject